# ADAMERGING: ADAPTIVE MODEL MERGING FOR MULTI-TASK LEARNING

**Enneng Yang**[1], **Zhenyi Wang**[2*], **Li Shen**[3*], **Shiwei Liu**[4], **Guibing Guo**[1*], **Xingwei Wang**[1], **Dacheng Tao**[5]

[1]Northeastern University, China [2]University of Maryland, USA [3]JD Explore Academy, China
[4]University of Oxford, UK [5]Nanyang Technological University, Singapore

ennengyang@stumail.neu.edu.cn, zwang169@umd.edu, mathshenli@gmail.com

shiwei.liu@maths.ox.ac.uk, {guogb,wangxw}@swc.neu.edu.cn, dacheng.tao@gmail.com

## ABSTRACT

Multi-task learning (MTL) aims to empower a model to tackle multiple tasks simultaneously. A recent development known as task arithmetic has revealed that several models, each fine-tuned for distinct tasks, can be directly merged into a single model to execute MTL without necessitating a retraining process using the initial training data. Nevertheless, this direct addition of models often leads to a significant deterioration in the overall performance of the merged model. This decline occurs due to potential conflicts and intricate correlations among the multiple tasks. Consequently, the challenge emerges of how to merge pre-trained models more effectively without using their original training data. This paper introduces an innovative technique called Adaptive Model Merging (`AdaMerging`). This approach aims to autonomously learn the coefficients for model merging, either in a task-wise or layer-wise manner, without relying on the original training data. Specifically, our `AdaMerging` method operates as an automatic, unsupervised task arithmetic scheme. It leverages entropy minimization on unlabeled test samples from the multi-task setup as a surrogate objective function to iteratively refine the merging coefficients of the multiple models. Our experimental findings across eight tasks demonstrate the efficacy of the AdaMerging scheme we put forth. Compared to the current state-of-the-art task arithmetic merging scheme, AdaMerging showcases a remarkable 11% improvement in performance. Notably, AdaMerging also exhibits superior generalization capabilities when applied to unseen downstream tasks. Furthermore, it displays a significantly enhanced robustness to data distribution shifts that may occur during the testing phase. The code is available at AdaMerging.

## 1 INTRODUCTION

Multi-task learning (MTL) is a technique that enables the transfer of knowledge (Wu et al., 2020; Wang et al., 2023; Jiang et al., 2024) among multiple tasks by efficiently sharing model parameters, leading to improvements in overall performance (Caruana, 1997; Liu et al., 2019b; Vandenhende et al., 2021) across a variety of tasks. Consequently, it has garnered significant attention in fields such as computer vision (Misra et al., 2016; Chen et al., 2018; 2020), natural language processing (Collobert & Weston, 2008; Dong et al., 2015), and recommendation systems (Ma et al., 2018; Yang et al., 2023; Song et al., 2024). In the context of foundation models, there are two key considerations. On the one hand, it is highly inefficient to pursue the traditional MTL approach for large pre-trained models by collecting a large volume of training data due to the high data labeling and computation cost. On the other hand, the advent of pre-trained models' popularity (Qiu et al., 2020) has led to a prevalent practice among downstream tasks. These tasks independently fine-tune the same pre-trained model, such as ViT (Dosovitskiy et al., 2021) or BERT (Devlin et al., 2019), and subsequently release these fine-tuned models, often without disclosing the specifics of their original training data. Consequently, there has emerged a recent trend in the research community, focused on exploring methodologies for effectively merging multiple independently trained models without relying on their training data for the purpose of MTL (Matena & Raffel, 2022; Jin et al., 2023; Ainsworth et al., 2023; Ilharco et al., 2023; Huang et al., 2023; Ortiz-Jimenez et al., 2023; Yadav et al., 2023; Li et al., 2023).

---

*Corresponding author

Recently, a novel concept in MTL known as task arithmetic has emerged (Ilharco et al., 2023). Task arithmetic introduces the notion of a "task vector", which can be described as a vector of weights fine-tuned specifically for a given task, subtracted from the corresponding pre-trained weights (as illustrated in Fig. 2(a)). Essentially, a task vector serves as a unique representation for a particular task. Research in this area, focusing on methods centered around task vectors (Ilharco et al., 2023; Yadav et al., 2023), has demonstrated that by summing multiple task vectors and integrating them into a pre-trained model, a new model can be created with the capability to handle multi-task learning effectively (as depicted in Fig. 2(b)). However, despite the promising results, there still exists a substantial performance gap between task vector-based MTL methods, such as Task Arithmetic (Ilharco et al., 2023) and Ties-Merging (Yadav et al., 2023), and traditional MTL approaches, as highlighted in Fig. 1. This disparity in performance suggests that further research and refinement are required to bridge the existing gap and unlock the full potential of task vector-based MTL methodologies.

A critical observation in the analysis of task vector-based MTL methods is the significance of the merging coefficient (denoted as $\lambda$ in Fig. 2(b)) associated with the task vector. This coefficient plays a pivotal role in determining the average accuracy of the final MTL model. As illustrated in Fig. 1, particularly in the cases of Task Arithmetic (indicated by the yellow line) and Ties-Merging (represented by the blue line), an ill-suited merging coefficient can lead to a situation where the model struggles to effectively perform MTL. In such scenarios, the average accuracy across multiple tasks becomes unacceptably low. This sensitivity to the merging coefficient may stem from potential conflicts (Guo et al., 2020; Vandenhende et al., 2021) or intricate relationships (Ma et al., 2018; Standley et al., 2020) among the multiple tasks, which make the merging process highly susceptible to the choice of this coefficient. Consequently, one of the primary challenges encountered in task vector-based MTL lies in determining the appropriate task vector merging coefficients that facili-

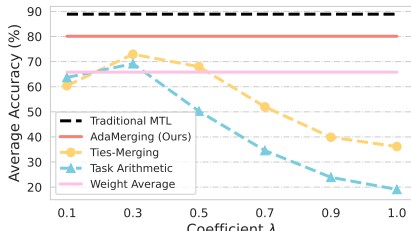

Figure 1: The impact of coefficient $\lambda$ on the average accuracy of various MTL methods on eight tasks. Among them, Task Arithmetic (Ilharco et al., 2023) and Ties-Merging (Yadav et al., 2023) based on task vectors achieved the best average accuracy when coefficient $\lambda = 0.3$, which were $69.1\%$ and $72.9\%$ respectively. Traditional MTL and our AdaMerging are $88.9\%$ and $80.1\%$.

tate the optimal integration of multiple tasks, all without relying on the original training data for each task. Additionally, it is more desirable and flexible to fine-tune different coefficients for different layers within the merged model. However, when dealing with a substantial number of tasks and layers, traditional approaches such as grid search (Liashchynskyi & Liashchynskyi, 2019) or combinatorial optimization search (Liu et al., 2020) become impractical for identifying suitable model merging coefficients. Hence, addressing this issue efficiently and effectively remains a challenging research problem in the field of task vector-based MTL.

In this paper, our inspiration comes from test-time adaptation schemes aimed at optimizing model generalization when faced with previously unseen test data (Wang et al., 2021; Liang et al., 2023). Building upon these concepts, we introduce an innovative automatic unsupervised multi-task model merging scheme. This scheme leverages the minimization of prediction distribution entropy on unlabeled multi-task test data as a surrogate objective to adaptively learn model merging coefficients. The intuitive motivation of entropy minimization is to make the model produce a more deterministic output when faced with a given input, which can lead to a more robust and accurate model. Our approach begins with an analysis of the relationship between entropy and prediction loss across eight tasks. As depicted in Fig. 3(a), our observations reveal that samples with lower entropy also exhibit smaller prediction losses. Furthermore, we calculate the Spearman correlation coefficient (Myers & Sirois, 2004) to quantify the relationship between entropy and prediction loss, as illustrated in Fig. 3(b). The results affirm a positive correlation between entropy and prediction loss, confirming that entropy can serve as a suitable proxy objective for optimization purposes. Subsequently, we put forth two adaptive model merging schemes, collectively referred to as `AdaMerging`. These schemes are designed to automatically learn a merging coefficient for each task vector or each layer of each task vector, as depicted in Fig. 2(c) and (d). To update these merging coefficients, we employ entropy minimization as a proxy objective, thereby enhancing the adaptability and performance of the multi-task model merging process.

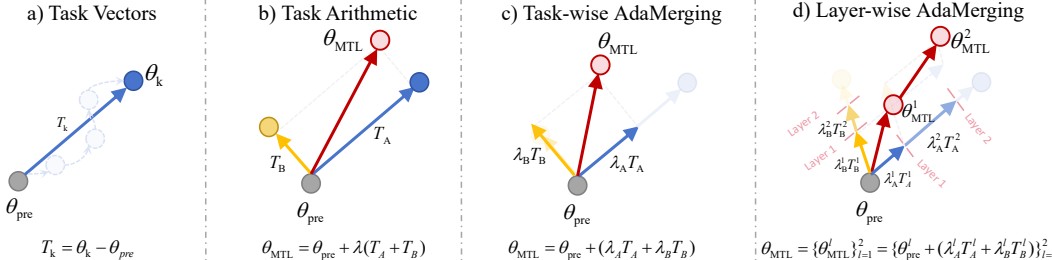

Figure 2: (a) Definition of "task vector", the task vector $T_k$ is obtained by subtracting the pre-trained weights $\theta_{pre}$ from the model weights $\theta_k$ fine-tuned on the data of task $k$. (b) *Task Arithmetic* (Ilharco et al., 2023) for MTL, which assigns same merging coefficient $\lambda$ to each task vector $T_k$ ($k \in \{A, B\}$). (c) `Task-wise AdaMerging` for MTL, which learns a distinct merging coefficient $\lambda_k$ to each task vector $T_k$ ($k \in \{A, B\}$). (d) `Layer-wise AdaMerging` for MTL, which learns a distinct merging coefficient $\lambda_k^l$ to each layer $l$ ($l \in \{1, 2\}$) of the task vector $T_k$ ($k \in \{A, B\}$).

Finally, we conduct a comprehensive evaluation to ascertain the superiority of AdaMerging when compared to existing task vector-based methods, revealing its advantages in three key aspects: (i) *Significantly Higher MTL Performance*: Our extensive testing across eight task vectors demonstrated that AdaMerging's adaptive learning merging coefficient significantly enhances the average accuracy across multiple tasks. For instance, on the ViT-B/32, AdaMerging improved approximately $5.0\%$ to $11.0\%$ over Task Arithmetic and Ties-Merging. (ii) *Substantially Improved Generalization*: Our evaluation on two sets of previously unseen downstream tasks underscored AdaMerging's superior generalization capabilities, resulting in improvements ranging from $4.4\%$ to $9.1\%$ when compared to Task Arithmetic and Ties-Merging. (iii) *Robust to Test Data Distribution Shifts*: In addition to performance gains, AdaMerging exhibited substantially enhanced robustness in multi-task testing across seven distribution drifts, with an average improvement of $8.45\%$ compared to Task Arithmetic.

This paper makes four significant **contributions**: (i) We re-examine existing task vector-based multi-task learning (MTL) methods and unveil the substantial influence of model merging coefficients on the average MTL performance. (ii) We introduce a novel approach called `AdaMerging`, which autonomously learns merging coefficients in an unsupervised manner. This method can adaptively determine coefficients for different task vectors (`Task-wise AdaMerging`) or individual layers within different task vectors (`Layer-wise AdaMerging`). (iii) We establish a strong positive correlation between entropy minimization and loss minimization on MTL's test data. This correlation signifies that these metrics can effectively serve as proxy objectives for optimizing the model merging coefficients within AdaMerging. (iv) We conduct comprehensive experiments to validate our method. The results demonstrate its substantial improvements in performance, generalization capabilities, and robustness compared to state-of-the-art (SOTA) task vector-based model merging methods.

## 2 RELATED WORK

**Joint Training for Multi-Task Learning**. The joint training method gathers training data from multiple tasks to learn these tasks simultaneously (Caruana, 1997) to achieve knowledge transfer (Wu et al., 2023). Existing works mainly focus on mitigating task conflicts from a architecture (Misra et al., 2016; Sun et al., 2020) or optimization (Sener & Koltun, 2018; Liu et al., 2021) perspective. Architectural-based methods mitigate task interference by sparsifying (Liu et al., 2019a; Ding et al., 2021), branching (Lu et al., 2017; Guo et al., 2020) or modularizing (Ma et al., 2018; Hazimeh et al., 2021) shared structures. Optimization-based methods balance multiple tasks from the perspectives of task training weights (Sener & Koltun, 2018; Liu et al., 2019a), gradient dominance (Chen et al., 2018; He et al., 2022; Yang et al., 2023), and gradient conflicts (Yu et al., 2020; Chen et al., 2020; Liu et al., 2021). However, the conventional approaches for collecting raw data across multiple tasks for joint training face challenges that may render them unsuitable in the era of foundation models. This is primarily due to either (i) their computational inefficiency stemming from the high computation cost for updating the pre-trained models or (ii) numerous data owners refrain from releasing valuable or privacy-sensitive raw data. Instead, they opt to share models fine-tuned on these pre-trained models.

**Model Merging for Multi-task Learning**. The practice of model merging has emerged as a promising solution to enhance model generalization and facilitate MTL. The first type of research

involves merging multiple models, all initially trained on the same task, with the aim of enhancing the model's overall generalization (Gupta et al., 2020; Cha et al., 2021; Wortsman et al., 2022; Wang et al., 2022) or to perform federated learning (Li et al., 2019; Wang et al., 2020; Liu et al., 2022). The other type of work attempts to merge models for different tasks to perform MTL (Matena & Raffel, 2022; Jin et al., 2023; Ainsworth et al., 2023; Stoica et al., 2023; Ortiz-Jimenez et al., 2023; Zhang et al., 2023; Ilharco et al., 2023; Yadav et al., 2023). This paper primarily concentrates on the latter approach. However, simple model averaging alone can significantly deteriorate performance across multiple tasks. Consequently, in recent years, numerous advanced techniques have surfaced to mitigate the performance loss associated with model merging. For example, Fisher Merging (Matena & Raffel, 2022) employs the Fisher information matrix (Fisher, 1922) to measure the importance of individual model parameter. Subsequently, it leverages this importance metric to guide the model merging. However, the computation of the Fisher information matrix becomes computationally and memory-intensive when dealing with a large number of model parameters. RegMean (Jin et al., 2023) suggests minimizing the $L_2$ distance between the merged model and each individual model. However, this approach necessitates the precomputation and provision of the inner product matrix for the training dataset. This information may not be accessible if the model owner chooses not to disclose it. In recent developments, Task Arithmetic (Ilharco et al., 2023), introduces the concept of "task vectors". This approach demonstrates that merging task vectors to create a consolidated model can effectively facilitate MTL. Building upon this foundation, PEM Composition (Zhang et al., 2023) extends the task arithmetic framework to incorporate the merging of LoRA (Hu et al., 2021) models. Taking this a step further, Ties-Merging (Yadav et al., 2023) addresses task conflicts within the Task Arithmetic paradigm. It accomplishes this by resetting redundant parameters, resolving sign conflicts, and exclusively merging parameters that exhibit sign-consistency. Task vector-based studies overlook a critical challenge encountered when dealing with a diverse collection of models, i.e., the coefficients governing the model merging process play a pivotal role in achieving optimal merging performance. In contrast, our work specifically emphasizes and addresses this issue to bridge the performance gap.

Overall, our work has three essential differences from existing task vector-based MTL schemes: (i) They share a merging coefficient across all task vectors, limiting the flexibility of task vector combinations. By contrast, our method adopts different merging coefficients across different tasks or even different layers, substantially enhancing the flexibility of adaptations. (ii) Existing works employ grid-searching the merging coefficients, thus lacking a guiding principle and is costly and infeasible when the number of tasks is large, while our work takes entropy minimization as a proxy objective to optimize the merging coefficients *efficiently and automatically*. (iii) We significantly improve multi-task performance, generalization to unseen tasks, and robustness to test data distribution shifts.

## 3 METHODOLOGY

We define the notation and model merging problem in Sec. 3.1, and briefly describe the solution based on task vectors. In Sec.3.2, we introduce the proposed `AdaMerging` method in detail.

### 3.1 PRELIMINARIES

**Notation**: Let $f_\theta(x_i) \to \hat{y}_i$ be a neural network model parameterized by a set of weights $\theta = \{\theta^1, \theta^2, \ldots, \theta^L\}$, which takes $x_i \in \mathbb{R}^d$ as an input data and outputs the predicted value $\hat{y}_i \in \mathbb{R}^C$. Among them, $\theta^l$ is the weight of the $l$-th ($l \in \{1, 2, \ldots, L\}$) layer, $L$ represents the number of layers of the network $f$, $d$ represents the dimension of the input data $x_i$, and $C$ represents the number of classes. Without loss of generality, we assume that the weights of a well-known pre-trained model, e.g., ViT (Dosovitskiy et al., 2021) or BERT (Devlin et al., 2019)), are $\theta_{pre} = \{\theta_{pre}^1, \theta_{pre}^2, \ldots, \theta_{pre}^L\}$.

There are $K$ tasks, and each of them has fine-tuned $\theta_{pre}$ on their own private training data $\{x_i, y_i\}_{i=1}^{N_k^{tr}}$, $N_k^{tr}$ represents the number of training samples for task $k$. Consequently, the model's weights after fine-tuning for task $k$ are recorded as $\theta_k = \{\theta_k^1, \theta_k^2, \ldots, \theta_k^L\}$.

**Problem Definition**: The *model merging* problem is defined as how to combine weights $\{\theta_k\}_{k=1}^K$ to get a new weight $\theta_{MTL}$ without necessitating a retraining process using the initial task's training data, and ensure that $f_{\theta_{MTL}}$ can perform tasks $1, 2, \ldots, K$ simultaneously. A straightforward approach is to perform weight averaging, i.e., $\theta_{MTL} = \frac{1}{K} \sum_{k=1}^K \theta_k$, however the performance of this approach usually drops dramatically (Ilharco et al., 2023; Yadav et al., 2023).

**Task Arithmetic**: A recent research (Ilharco et al., 2023) defines the concept of "task vectors" and completes various task arithmetic operations based on task vectors, such as *adding* multiple task vectors to the pre-trained weight $\theta_{pre}$ to perform MTL. Specifically, as shown in Fig. 2(a), the task vector $T_k$ w.r.t task $k$ is defined as a vector obtained by performing a subtraction operation with the fine-tuned weights $\theta_k$ and the pre-trained weights $\theta_{pre}$, i.e., $T_k = \theta_k - \theta_{pre}$. Furthermore, multiple task vectors $\{T_k\}_{k=1}^K$ are added and merged into the pre-trained model, $\theta_{MTL} = \theta_{pre} + \lambda \sum_{k=1}^K T_k$, where the coefficient $\lambda$ represents the importance of model merging. On this basis, Ties-Merging (Yadav et al., 2023) shows that some parameter values in the task vector may be redundant, or the signs of the parameters may conflict, and direct merging will cause performance losses. Based on this assumption, they proposed to perform three steps of *Trim, Elect Sign and Disjoint Merge* on merging task vectors. We combine these steps and abbreviate them as one $\Phi()$ operation. Therefore, model merging in Ties-Merging can be expressed as $\theta_{MTL} = \theta_{pre} + \lambda \sum_{k=1}^K \Phi(T_k)$.

Task arithmetic is a simple and effective idea. As shown in Fig. 1, task vectors based MTL model merging methods, i.e., Task Arithmetic (blue line), Ties-Merging (yellow line), are significantly better than simple weight averaging scheme (pink line). However, there is still a clear gap between them and the traditional MTL (black line). In addition, task vector-based model merging methods are very sensitive to the merging coefficient $\lambda$. An ill-suited $\lambda$ will cause the performance to be lower than the weighted average, or even reach unacceptably low accuracy. When the number of tasks is large, grid searching the merging coefficients for each task vector is expensive. This motivates us to conduct further research to narrow the performance gap between task vector-based MTL and traditional MTL.

## 3.2 ADAPTIVE MODEL MERGING FOR MULTI-TASK LEARNING

In this section, we propose an *unsupervised* adaptive model merging method for task vectors based MTL, called `AdaMerging`. It makes the merging coefficient of each task vector learnable (`Task-wise AdaMerging`). Furthermore, different layers of a task vector can also automatically learn different merging coefficients in AdaMerging (`Layer-wise AdaMerging`).

### 3.2.1 ADAMERGING: ADAPTIVE MODEL MERGING

**Task-wise AdaMerging**: As shown in Fig. 2(c), our Task-wise AdaMerging assigns a separate merging coefficient $\lambda_k$ to each task vector $T_k$, that is: $\theta_{MTL} = \theta_{\text{pre}} + \sum_{k=1}^K \lambda_k T_k$. Task-wise AdaMerging allows task vectors $T_k$ that have a positive transfer to the average MTL performance to occupy a larger proportion in $\theta_{MTL}$, while task vector $T_{k'}$ that is harmful to MTL will reduce their contribution to the merging weight $\theta_{MTL}$, thereby improving the average MTL performance.

**Layer-wise AdaMerging**: However, Task-wise AdaMerging may not be enough to alleviate the interference of task vectors. In the deep neural network model, the information learned by each layer is different. For example, the lower layer may learn general features, while the higher layers may learn task-specific features (Yosinski et al., 2014). Therefore, when merging task vectors, the weights $\{T_k^1, T_k^2, \ldots, T_k^L\}$ of different layers for each task vector $T_k$ should also have different contributions $\{\lambda_k^1, \lambda_k^2, \ldots, \lambda_k^L\}$ to the final multi-task weights $\theta_{MTL}$. Based on this, we propose the Layer-wise AdaMerging scheme shown in Fig. 2(d), which is formalized as: $\theta_{MTL} = \{\theta_{MTL}^l\}_{l=1}^L = \{\theta_{\text{pre}}^l + \sum_{k=1}^K \lambda_k^l T_k^l\}_{l=1}^L$, where $L$ represents the number of layers.

**AdaMerging++**: The above AdaMerging adaptively merges the original task vector $T_k$ in the Task Arithmetic (Ilharco et al., 2023). Naturally, it can also adaptively merge the task vector $\Phi(T_k)$ after removing parameter redundant values and sign conflicts in Ties-Merging (Yadav et al., 2023). We call this variant AdaMerging++, and the corresponding Task-wise AdaMerging++ and Layer-wise AdaMerging++ versions are formalized as $\theta_{MTL} = \theta_{\text{pre}} + \sum_{k=1}^K \lambda_k \Phi(T_k)$ and $\theta_{MTL} = \{\theta_{MTL}^l\}_{l=1}^L = \{\theta_{\text{pre}}^l + \sum_{k=1}^K \lambda_k^l \Phi(T_k^l)\}_{l=1}^L$, respectively.

Now, AdaMerging/AdaMerging++ faces a critical challenge, that is, we only have the task vector of each task *without their initial training data*. How to optimize merging coefficients $\{\lambda_k\}_{k=1}^K$ (or $\{\lambda_k^l\}_{k=1,l=1}^{K,L}$)? Our inspiration to solve this challenge comes from test-time adaptation (Wang et al.,

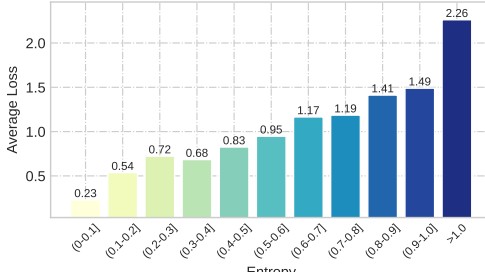 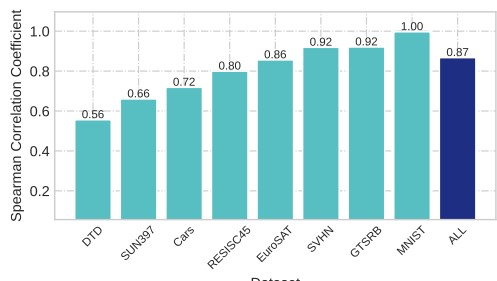

Figure 3: Correlation of *entropy* $H(\hat{Y})$ and *avareage loss* $L(Y, \hat{Y})$ on eight tasks (or datasets). (a) We divided the test samples into eleven groups according to the entropy of the samples, and observed the average prediction loss of the samples in each group. We observe that groups with smaller entropy correspond to smaller average losses. (b) We calculated the Spearman correlation coefficient between entropy and prediction loss on eight tasks (or datasets) and observed a high positive correlation.

2021; Niu et al., 2022; 2023), they adapt the weights of the trained model on unseen test data to cope with the distribution shifts on the test data.

### 3.2.2 ENTROPY OPTIMIZATION

We use *entropy minimization* on multi-task unlabeled test samples as an optimization surrogate objective function to update the merging coefficients $\{\lambda_k\}_{k=1}^K$ (or $\{\lambda_k^l\}_{k=1,l=1}^{K,L}$) in our AdaMerging.

**Entropy Minimization**: For a sample $x_i$, the predicted output of a neural network $f_\theta(x_i)$ is $\hat{y}_i$, the corresponding Shannon entropy (Shannon, 1948) is expressed as $H(\hat{y}_i) = -\sum_c^C p(\hat{y}_{i,c}) \log p(\hat{y}_{i,c})$, where $p(\hat{y}_{i,c}) \in [0,1]$ represents the probability that the input $x_i$ is predicted to be the $c$-th class. Previous research on test-time adaptation (Wang et al., 2021; Niu et al., 2023) found that optimizing the model's parameters based on entropy minimization (Grandvalet & Bengio, 2004; Roy et al., 2022), $\min H(\hat{y}_i)$, on test samples can make the model adapt to unseen test data distributions.

However, it is unclear whether entropy minimization can be used as an effective surrogate objective function in multi-task model merging. To verify whether entropy minimization can be used as a proxy objective for MTL loss, we performed the analysis on the eight tasks used in the experiment. First, we combine the *test data* of the eight tasks as $(X, Y) = \{\{x_i, y_i\}_{i=1}^{N_k^{te}}\}_{k=1}^K$, and compute the prediction of the multi-task model $f_{\theta_{MTL}}$ on test data as $\hat{Y} = \{\{f_{\theta_{MTL}}(x_i)\}_{i=1}^{N_k^{te}}\}_{k=1}^K$. Next, we calculate the loss between the real label $Y$ and the predicted value $\hat{Y}$ case-by-case and obtain $L(Y, \hat{Y}) = \{\ell(y_i, \hat{y}_i)\}_{i=1}^{|\hat{Y}|}$, where $|\hat{Y}|$ represents the total number of test samples for all tasks, and $\ell$ represents a loss function, such as cross-entropy. We also calculate the entropy of each sample on the test set and get $H(\hat{Y}) = \{H(\hat{y}_i)\}_{i=1}^{|\hat{Y}|}$. Finally, we analyze the correlation between entropy $H(\hat{Y})$ and prediction loss $L(Y, \hat{Y})$ from two aspects. (i) We divide the multi-task samples into multiple intervals based on entropy $H(\hat{Y})$ from small to large, such as $\mathcal{I} = \{\mathcal{I}_1, \mathcal{I}_2, \ldots, \mathcal{I}_{11}\} = \{(0.0, 0.1], (0.1 - 0.2], \ldots, (1.0, \infty)\}$, and count the average prediction loss of the samples contained in each interval $\mathcal{I}_t$ ($t \in \{1, 2, \ldots, 11\}$). As shown in Fig. 3(a), we observe that the average loss corresponding to the interval with small entropy is also smaller. (ii) We also directly calculated the Spearman correlation coefficient (Myers & Sirois, 2004) of entropy $H(\hat{Y})$ and prediction loss $L(Y, \hat{Y})$. As shown in Fig. 3, we observe that the average correlation between the two on multi-task data (i.e., dark purple "ALL") is as high as $0.87$. Therefore, we can conclude that entropy minimization (i.e., $\min H(\hat{Y})$) can serve as an effective surrogate objective for loss minimization (i.e., $\min L(Y, \hat{Y})$) on MTL. In Fig. 10 of the Appendix, we further verify that this correlation exists across different training stages of model merging.

**Optimization Objective**: Based on the above verification, we take entropy minimization as the optimization proxy goal of the model merging coefficient in our AdaMerging/AdaMerging++. For example, the optimization form of the merging coefficient in Task-wise AdaMerging is:

$$\min_{\lambda_1, \lambda_2, \ldots, \lambda_K} \sum_{k=1}^K \sum_{x_i \in \mathcal{B}_k} H(f_{\theta_{MTL}}(x_i)), \text{ where } \theta_{MTL} = \theta_{\text{pre}} + \sum_{k=1}^K \lambda_k T_k,$$

where $\mathcal{B}_k$ represents a batch of unlabeled test samples sampled in task $k$. The coefficient $\{\lambda_k\}_{k=1}^K$ can be updated iteratively by obtaining the gradient through backpropagation. This is trivial with automatic differentiation tools like Pytorch (Paszke et al., 2017). It should be emphasized that, on the one hand, we do not need all test data to be available. Even if only $0.1\%$ or $1\%$ of unlabeled tests are available, our method can have significant performance improvements. On the other hand, our extra training time is also very cheap. These results are presented in the appendix.

# 4 EXPERIMENT

In this section, we introduce the experimental setup in Sec. 4.1 and the experimental results in Sec. 4.2. Due to page limitations, some details and results are shown in the **Appendix**.

## 4.1 EXPERIMENT SETUP

**Datasets and Models**: Following  Ilharco et al. (2023) and  Yadav et al. (2023), we study task vectors based multi-task model merging on eight image classification datasets: SUN397 (Xiao et al., 2016), Cars (Krause et al., 2013), RESISC45 (Cheng et al., 2017), EuroSAT (Helber et al., 2019), SVHN (Yuval, 2011), GTSRB (Stallkamp et al., 2011), MNIST (LeCun, 1998), DTD (Cimpoi et al., 2014). We provide a more detailed description of the dataset in the Appendix A. In the main text, we use the Vit-B/32 and ViT-L/14 architectures in CLIP (Radford et al., 2021) as pre-trained models to conduct experiments. We also report the results on the Vit-B/16 architecture in the Appendix B.

**Baselines and Metric**: Our baselines are mainly divided into two categories, one is non-model merging, i.e., Individual and Traditional MTL; and the other is various advanced model merging methods, such as Weight Averaging, Fisher Merging (Matena & Raffel, 2022), RegMean (Jin et al., 2023), Task Arithmetic (Ilharco et al., 2023) and Ties-Merging (Yadav et al., 2023). Baseline details are provided in Appendix A. Among them, *Task Arithmetic* and *Ties-Merging* are task vectors based MTL methods, which are also our most important baselines. In addition, our methods include Task-wise AdaMerging, Task-wise AdaMerging++, Layer-wise AdaMerging, and Layer-wise AdaMerging++. Unless otherwise specified, our method uses the *Layer-wise* version. We report the **average accuracy** (i.e., Avg Acc) of MTL model on the test set of all tasks as an evaluation metric.

## 4.2 PERFORMANCE, GENERALIZATION, ROBUSTNESS

In this section, we demonstrate the superiority of our approach over SOTA methods for merging task vectors by evaluating it from three key perspectives: performance, generalization and robustness.

**Significantly Higher MTL Performance**. We verify that the proposed AdaMerging method significantly outperforms existing model merging methods in performance. As shown in Tab. 1 and Tab. 2, we tested the performance of merging ViT-B/32 and ViT-L/14 on eight tasks, respectively. We have the following observations: (i) Individual and Traditional MTL methods achieve the optimal performance, which are 90.5% and 88.9% under ViT-B/32. However, they all rely on initial training data for multiple tasks. Additionally, independent fine-tuning requires storing a model for each task. (ii) Weight Averaging is the simplest model merging solution. Naturally, its performance is also the lowest. Furthermore, Fisher Merging merged models by calculating parameter importance, and RegMean imposed the constraint that the distance between the merged MTL model and a single model is close. Both of them perform better compared to the Weight Averaging. (iii) Advanced task vectors

Table 1: Multi-task performance when merging ViT-B/32 models on eight tasks.

| Method | SUN397 | Cars | RESISC45 | EuroSAT | SVHN | GTSRB | MNIST | DTD | Avg Acc |
|---|---|---|---|---|---|---|---|---|---|
| Pretrained | 62.3 | 59.7 | 60.7 | 45.5 | 31.4 | 32.6 | 48.5 | 43.8 | 48.0 |
| Individual | 75.3 | 77.7 | 96.1 | 99.7 | 97.5 | 98.7 | 99.7 | 79.4 | 90.5 |
| Traditional MTL | 73.9 | 74.4 | 93.9 | 98.2 | 95.8 | 98.9 | 99.5 | 77.9 | 88.9 |
| Weight Averaging | 65.3 | 63.4 | 71.4 | 71.7 | 64.2 | 52.8 | 87.5 | 50.1 | 65.8 |
| Fisher Merging (Matena & Raffel, 2022) | 68.6 | 69.2 | 70.7 | 66.4 | 72.9 | 51.1 | 87.9 | 59.9 | 68.3 |
| RegMean (Jin et al., 2023) | 65.3 | 63.5 | 75.6 | 78.6 | 78.1 | 67.4 | 93.7 | 52.0 | 71.8 |
| Task Arithmetic (Ilharco et al., 2023) | 55.2 | 54.9 | 66.7 | 78.9 | 80.2 | 69.7 | 97.3 | 50.4 | 69.1 |
| Ties-Merging (Yadav et al., 2023) | 59.8 | 58.6 | 70.7 | 79.7 | 86.2 | 72.1 | 98.3 | 54.2 | 72.4 |
| **Task-wise AdaMerging** (Ours) | 58.0 | 53.2 | 68.8 | 85.7 | 81.1 | 84.4 | 92.4 | 44.8 | 71.1 |
| **Task-wise AdaMerging++** (Ours) | 60.8 | 56.9 | 73.1 | 83.4 | 87.3 | 82.4 | 95.7 | 50.1 | 73.7 |
| **Layer-wise AdaMerging** (Ours) | 64.5 | 68.1 | 79.2 | 93.8 | 87.0 | 91.9 | 97.5 | 59.1 | 80.1 |
| **Layer-wise AdaMerging++** (Ours) | 66.6 | 68.3 | 82.2 | 94.2 | 89.6 | 89.0 | 98.3 | 60.6 | 81.1 |

Table 2: Multi-task performance when merging ViT-L/14 models on eight tasks.

| Method | SUN397 | Cars | RESISC45 | EuroSAT | SVHN | GTSRB | MNIST | DTD | Avg Acc |
|---|---|---|---|---|---|---|---|---|---|
| Pretrained | 66.8 | 77.7 | 71.0 | 59.9 | 58.4 | 50.5 | 76.3 | 55.3 | 64.5 |
| Individual | 82.3 | 92.4 | 97.4 | 100 | 98.1 | 99.2 | 99.7 | 84.1 | 94.2 |
| Traditional MTL | 80.8 | 90.6 | 96.3 | 96.3 | 97.6 | 99.1 | 99.6 | 84.4 | 93.5 |
| Weight Averaging | 72.1 | 81.6 | 82.6 | 91.9 | 78.2 | 70.7 | 97.1 | 62.8 | 79.6 |
| Fisher Merging (Matena & Raffel, 2022) | 69.2 | 88.6 | 87.5 | 93.5 | 80.6 | 74.8 | 93.3 | 70.0 | 82.2 |
| RegMean (Jin et al., 2023) | 73.3 | 81.8 | 86.1 | 97.0 | 88.0 | 84.2 | 98.5 | 60.8 | 83.7 |
| Task Arithmetic (Ilharco et al., 2023) | 73.9 | 82.1 | 86.6 | 94.1 | 87.9 | 86.7 | 98.9 | 65.6 | 84.5 |
| Ties-Merging (Yadav et al., 2023) | 76.5 | 85.0 | 89.3 | 95.7 | 90.3 | 83.3 | 99.0 | 68.8 | 86.0 |
| **AdaMerging** (Ours) | 79.0 | 90.3 | 90.8 | 96.2 | 93.4 | 98.0 | 99.0 | 79.9 | 90.8 |
| **AdaMerging++** (Ours) | 79.4 | 90.3 | 91.6 | 97.4 | 93.4 | 97.5 | 99.0 | 79.2 | 91.0 |

Table 3: Generalization results on two unseen tasks when merging ViT-B/32 models on six tasks.

| Method | Seen Tasks | | | | | | | Unseen Tasks | | |
|---|---|---|---|---|---|---|---|---|---|---|
| | SUN397 | Cars | RESISC45 | DTD | SVHN | GTSRB | Avg Acc | MNIST | EuroSAT | Avg Acc |
| Task Arithmetic (Ilharco et al., 2023) | 63.3 | 62.4 | 75.1 | 57.8 | 84.6 | 80.4 | 70.6 | 77.2 | 46.2 | 61.7 |
| Ties-Merging (Yadav et al., 2023) | 67.8 | 66.2 | 77.2 | 56.7 | 77.1 | 70.9 | 69.3 | 75.9 | 43.3 | 59.6 |
| **AdaMerging** (Ours) | 65.2 | 65.9 | 88.5 | 61.1 | 92.2 | 91.5 | 77.4 | 84.0 | 56.1 | 70.0 |
| **AdaMerging++** (Ours) | 68.2 | 67.6 | 86.3 | 63.6 | 92.6 | 89.8 | 78.0 | 83.9 | 53.5 | 68.7 |

| Method | SUN397 | Cars | GTSRB | EuroSAT | DTD | MNIST | Avg Acc | RESISC45 | SVHN | Avg Acc |
|---|---|---|---|---|---|---|---|---|---|---|
| Task Arithmetic (Ilharco et al., 2023) | 64.0 | 64.0 | 75.2 | 87.7 | 57.0 | 95.7 | 73.9 | 52.3 | 44.9 | 51.1 |
| Ties-Merging (Yadav et al., 2023) | 68.0 | 67.1 | 67.7 | 78.4 | 56.5 | 92.8 | 71.8 | 58.7 | 49.2 | 53.9 |
| **AdaMerging** (Ours) | 67.1 | 67.8 | 94.8 | 94.4 | 59.6 | 98.2 | 80.3 | 50.2 | 60.9 | 55.5 |
| **AdaMerging++** (Ours) | 68.9 | 69.6 | 91.6 | 94.3 | 61.9 | 98.7 | 80.8 | 52.0 | 64.9 | 58.5 |

based multi-task merging methods (i.e., Task Arithmetic and Ties-Merging) have achieved good performance. For example, Ties-Merging has achieved the performance in ViT-B/32 and ViT-L/14 by 72.4% and 86.0%, respectively. However, there is still a big gap between this and Traditional MTL (i.e., 88.9% and 93.5%, respectively). (iv) Our Task-wise AdaMerging and Task-wise AdaMerging++ use unsupervised learnable coefficients to merge task vectors in Task Arithmetic and Ties-Merging respectively, bringing 2% and 1.3% performance improvements respectively on ViT-B/32. Thanks to the more fine-grained fusion solution, on ViT-B/32, our Layer-wise AdaMerging and Layer-wise AdaMerging++ bring 11% and 8.7% performance improvements compared to Task Arithmetic and Ties-Merging, while on ViT-L/14, our method brought improvements of 6.3% and 5.0%. Our AdaMerging greatly reduces the gap between model merging and traditional MTL solutions.

**Substantially Improved Generalization**. MTL hopes to transfer the knowledge of old tasks to new tasks and improve the generalization of the MTL model. To this end, we compare the performance of AdaMerging and task vector-based model merging methods (Task Arithmetic and Ties-Merging) on two sets of unseen tasks. In Tab. 3, we merge the task vectors corresponding to six tasks and test on two unseen tasks (i.e. their task vectors are not merged). We observe: (i) On the six seen tasks, AdaMerging and AdaMerging++ are significantly better than Task Arithmetic and Ties-Merging. (ii) More importantly, AdaMerging method maintains this superiority on two unseen tasks. For example, on the two tasks of MNIST and EuroSAT, the average performance of AdaMerging and AdaMerging++ improved by 8.3% and 9.1%, respectively, compared with Task Arithmetic and Ties-Merging. In addition, on the two unseen tasks of RESISC45 and SVHN, the average accuracy improvements of AdaMerging and AdaMerging++ are 4.4% and 5.4%, respectively. These results indicate that our AdaMerging and AdaMerging++ methods generalize better to unseen tasks.

**Robust to Test Data Distribution Shifts**. Considering that the model provider only releases the fine-tuned model and does not expose the original training data, the model merger's test data may differ from the model owner's training data. we tested whether AdaMerging is still effective when the test data distribution shifts significantly. Following Hendrycks & Dietterich (2019), we created 7 corruption test data, and examples of corrupted images are shown Fig. 5 in Appendix B. The results on ViT-B/32 are shown in Tab. 4. On clean test data, AdaMerging has an 8.2% performance improvement compared to Task Arithmetic. On the corruption test datasets of Motion Blur, Impulse Noise, Gaussian Noise, Pixelate, Spatter, Contrast and JPEG Compression, AdaMerging's performance is 11.2%, 6.7%, 5.8%, 8.9%, 6.7%, 10.1% and 9.8% higher than Task Arithmetic respectively. These evidences fully demonstrate that our AdaMerging is more robust to test data distribution shifts.

**Summary**: Our AdaMerging/AdaMerging++ allows us to adapt to unlabeled test data of task vectors, unlabeled test data of unseen tasks, or unlabeled corruption data in an unsupervised way when training model merging coefficients, thereby optimizing the best suitable model merging coefficients are used to obtain a model with better performance, generalization or robustness.

Table 4: Robustness results when merging ViT-B/32 models on four tasks.

| Method | Clean Test Set | | | | | Corruption Test Set (Motion Blur ) | | | | |
|---|---|---|---|---|---|---|---|---|---|---|
| | Cars | EuroSAT | RESISC45 | GTSRB | Avg Acc | Cars | EuroSAT | RESISC45 | GTSRB | Avg Acc |
| Task Arithmetic | 66.9 | 94.7 | 82.6 | 75.1 | 79.8 | 65.3 | 68.1 | 80.0 | 64.2 | 69.4 |
| **AdaMerging** (Ours) | 73.7 | 96.1 | 85.8 | 96.3 | 88.0 | 71.2 | 74.6 | 82.7 | 94.1 | 80.6 |

| Method | Corruption Test Set (Impulse Noise) | | | | | Corruption Test Set (Gaussian Noise) | | | | |
|---|---|---|---|---|---|---|---|---|---|---|
| | Cars | EuroSAT | RESISC45 | GTSRB | Avg Acc | Cars | EuroSAT | RESISC45 | GTSRB | Avg Acc |
| Task Arithmetic | 62.1 | 49.1 | 72.7 | 40.4 | 56.1 | 63.6 | 55.4 | 75.9 | 49.4 | 61.1 |
| **AdaMerging** (Ours) | 67.2 | 30.8 | 75.9 | 77.5 | 62.8 | 69.9 | 41.2 | 80.6 | 76.0 | 66.9 |

| Method | Corruption Test Set (Pixelate) | | | | | Corruption Test Set (Spatter) | | | | |
|---|---|---|---|---|---|---|---|---|---|---|
| | Cars | EuroSAT | RESISC45 | GTSRB | Avg Acc | Cars | EuroSAT | RESISC45 | GTSRB | Avg Acc |
| Task Arithmetic | 2.78 | 41.5 | 22.8 | 66.6 | 33.4 | 63.3 | 60.1 | 73.9 | 54.3 | 62.9 |
| **AdaMerging** (Ours) | 2.49 | 53.8 | 22.4 | 90.6 | 42.3 | 69.9 | 43.6 | 75.4 | 89.4 | 69.6 |

| Method | Corruption Test Set (Contrast) | | | | | Corruption Test Set (JPEG Compression) | | | | |
|---|---|---|---|---|---|---|---|---|---|---|
| | Cars | EuroSAT | RESISC45 | GTSRB | Avg Acc | Cars | EuroSAT | RESISC45 | GTSRB | Avg Acc |
| Task Arithmetic | 66.0 | 62.9 | 75.9 | 70.6 | 68.9 | 66.5 | 72.3 | 82.2 | 60.0 | 70.3 |
| **AdaMerging** (Ours) | 71.7 | 69.8 | 79.3 | 95.1 | 79.0 | 70.9 | 75.8 | 83.6 | 90.1 | 80.1 |

### 4.3 ADAMERGING ANALYSIS

**Task-wise Coefficients**. In Tab. 5, we consistently observe that the merging coefficients of each task vector are inconsistent. When the number of tasks is relatively large, it is obviously undesirable to grid search the coefficients of each task, but our AdaMerging avoids this manual search process.

Table 5: Model merging coefficients $\{\lambda_k\}_{k=1}^{K}$ change with respect to training steps on ViT-B/32.

| Method | SUN397 | Cars | RESISC45 | EuroSAT | SVHN | GTSRB | MNIST | DTD |
|---|---|---|---|---|---|---|---|---|
| Task-wise AdaMerging | 0.2202 | 0.1413 | 0.2826 | 0.3284 | 0.2841 | 0.4003 | 0.1978 | 0.1692 |
| Task-wise AdaMerging++ | 0.3171 | 0.1698 | 0.4235 | 0.5198 | 0.4386 | 0.5803 | 0.2452 | 0.2885 |

**Layer-wise Coefficients**. Fig. 4 shows the merging coefficients learned by Layer-wise AdaMerging and AdaMerging++ on ViT-B/32 respectively. We observed that: (i) The coefficients learned by each layer of each task vector are different, which shows that the importance of each layer in the model merging process is different. (ii) The coefficients learned by shallow layers are generally smaller than those of deep layers, which indicates that shallow layers rely more on the weights of the pre-trained model rather than the weights provided by task vectors, while the deep layers rely more on the weights provided by the task vectors. This may be since the shallow layer learns general features, which are cross-task, while the deep layer learns task-specific features (Yosinski et al., 2014).

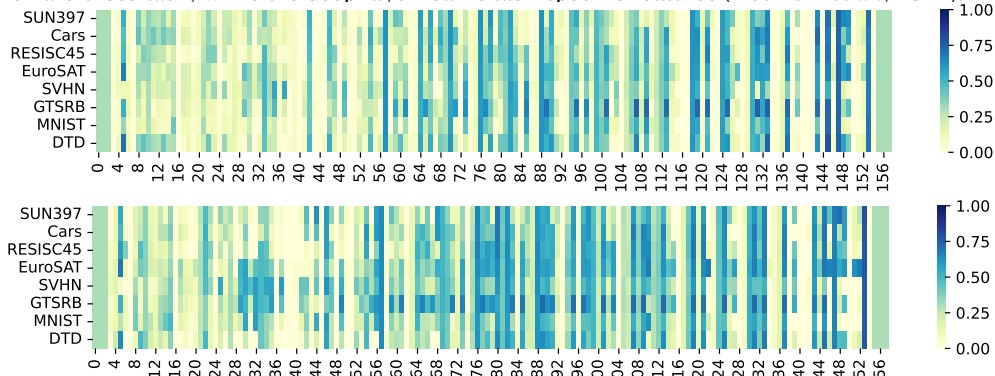

Figure 4: Learned model merging coefficients $\{\lambda_k^l\}_{k=1,l=1}^{K,L}$ of Layer-wise AdaMerging (Above) and AdaMerging++ (Below) on ViT-B/32. The $k$-th row represents the $k$-th task vector, the $l$-th column represents the $l$-th layer, and the intersection point represents the coefficient $\lambda_k^l$.

## 5 CONCLUSION AND FUTURE WORK

Advanced task arithmetic shows that new models built by merging multiple task vectors into a pre-trained model can execute MTL without needing original training data. However, task vector-based MTL methods are very sensitive to the merging coefficient. In this paper, we propose an adaptive model merging scheme (abbreviated as `AdaMerging`) to solve this problem, which takes entropy minimization as a surrogate objective to automatically learn the merging coefficients for each task vector or layer. Experimental results show that the proposed AdaMerging is superior to the current SOTA model merging methods in multi-task performance, generalization and robustness. In the future, we plan to further explore model merging solutions for different architectures.

ACKNOWLEDGMENTS

Li Shen is supported by STI 2030—Major Projects (No. 2021ZD0201405). Enneng Yang and Guibing Guo are supported by the National Natural Science Foundation of China under Grant No. 62032013, the Science and technology projects in Liaoning Province (No. 2023JH3/10200005), and the Fundamental Research Funds for the Central Universities under Grant No. N2317002.

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

## A    EXPERIMENT SETTINGS

This section provides a detailed dataset description, baseline description, and training details.

**Dataset Details**. Following Task Arithmetic (Ilharco et al., 2023), Ties-Merging (Yadav et al., 2023), we study multi-task model merging on eight image classification datasets below.

- **SUN397** (Xiao et al., 2016) is a scene classification dataset, which contains images in 397 classes, with a total of 108,754 images, and each class has at least 100 images.
- **Stanford Cars (Cars)** (Krause et al., 2013) is a car classification dataset, which contains 196 classes of cars and a total of 16,185 images. Each class in the training set and test set is divided at a ratio of 1:1.
- **RESISC45** (Cheng et al., 2017) is a remote sensing image scene classification data set. It contains 45 classes of scenes and a total of 31,500 images, of which there are approximately 700 images in each class.
- **EuroSAT** (Helber et al., 2019) is a satellite image classification dataset containing 27,000 labeled and geo-referenced images in 10 classes.
- **SVHN** (Yuval, 2011) is a real-world digital classification data set extracted from house numbers in Google Street View images. There are 10 classes in total. The training set contains 73,257 samples, the test set contains 26,032 samples, and 531,131 additional simple samples can be used as additional training data.
- **GTSRB** (Stallkamp et al., 2011) is a traffic sign classification dataset, which contains 43 classes of traffic signs with a total sample size of more than 50,000.
- **MNIST** (LeCun, 1998) is a benchmark dataset for image classification. It contains grayscale images of handwritten digits in 10 classes. The number of images in the training and test sets is 60,000 and 10,000 respectively. The number of images in each class is balanced.
- **DTD** (Cimpoi et al., 2014) is a texture classification data set, which contains 47 classes, a total of 5,640 images, and each class has approximately 120 images.

**Baseline Details**.  Our experiments involve the following seven comparison methods and four variations of our method.

- **Individual** means that each task uses an independent fine-tuned model, which has no interference between tasks, but cannot perform multiple tasks simultaneously.
- **Traditional MTL** collects the original training data of all tasks together to train a multi-task model. It can be used as a reference *upper bound* for model merging work.
- **Weight Averaging** is the simplest method of model merging, which directly averages the parameters of multiple models. It can be used as a *lower bound* for model merging.
- **Fisher Merging** (Matena & Raffel, 2022) calculates the Fisher information matrix (Fisher, 1922) to measure the importance of each parameter when merging models, and model merging is performed according to the guidance of this importance.
- **RegMean** (Jin et al., 2023) imposes a constraint when merging models, that is, the $L_2$ distance between the merged model and a single model is required to be as small as possible.
- **Task Arithmetic** (Ilharco et al., 2023) first defines the concept of "task vectors" and merges task vectors into a pre-trained model to execute multi-task learning.
- **Ties-Merging** (Yadav et al., 2023) further solves the task conflict problem in Task Arithmetic (Ilharco et al., 2023). It eliminates redundant parameters and resolves symbol conflicts through three steps: Trim, Elect Sign, and Disjoint Merge.
- **Task-wise AdaMerging (Ours)** is based on Task Arithmetic (Ilharco et al., 2023), which uses an unsupervised method to automatically learn the merging coefficient of the task vector in Task Arithmetic.
- **Task-wise AdaMergign++ (Ours)** is based on Ties-Merging (Yadav et al., 2023), which uses an unsupervised approach to learn a merging coefficient for each task vector in Ties-Merging.
- **Layer-wise AdaMerging (Ours)** automatically learns a merging coefficient for each layer of each task vector in Task Arithmetic (Ilharco et al., 2023).
- **Layer-wise AdaMergign++ (Ours)** uses an unsupervised approach to learn a merging coefficient for each layer of each task vector in Ties-Merging (Yadav et al., 2023).

**Implementation Details**. For the seven baseline methods, we follow the experimental settings in Task Arithmetic (Ilharco et al., 2023) and Ties-Merging (Yadav et al., 2023).  In our experiments,

the merging coefficient $\lambda$ of Task Arithmetic and Ties-Merging is set to 0.3 by default. For our four variants, we initialize all coefficients $\{\lambda_k\}_{k=1}^{K}$ (or $\{\lambda_k^l\}_{k=1,l=1}^{K,L}$) to 0.3 by default before learning and then update them unsupervised. We use an Adam optimizer (Kingma & Ba, 2014) to update the merging coefficients, with the learning rate set to 0.001, the momentum to (0.9, 0.999), and the batch size to 16. To avoid significantly increasing training costs, we only trained 500 iterations to update the merging coefficient. Pre-trained models ViT-B/32, ViT-B/16 and ViT-L/14 from CLIP (Radford et al., 2021) like Task Arithmetic (Ilharco et al., 2023) and Ties-Merging (Yadav et al., 2023).

# B  EXPERIMENT RESULTS

## B.1  PERFORMANCE, GENERALIZATION AND ROBUSTNESS

**Performance**. Tab. 6 shows the average accuracy of merging ViT-B/16 on eight tasks. We can observe that: (i) Ties-Merging alleviates the conflict problem of task vectors in Task Arithmetic, thus achieving a 3.2% performance improvement compared to Task Arithmetic. (ii) Our Task-wise AdaMerging and AdaMerging++ automatically learn a merging coefficient for each task vector in Task Arithmetic and Ties-Merging, thus bringing about 2.2% and 1.0% performance improvements, respectively. (iii) Our Layer-wise AdaMerging and AdaMerging++ further adaptively learn a merging coefficient for each layer of each task vector in Task Arithmetic and Ties-Merging, ultimately achieving performance improvements of 11.1% and 8.7%. These results further demonstrate the effectiveness of our AdaMerging scheme in multi-task model merging.

Table 6: Multi-task performance when merging ViT-B/16 models on eight tasks.

| Method | SUN397 | Cars | RESISC45 | EuroSAT | SVHN | GTSRB | MNIST | DTD | Avg Acc |
|---|---|---|---|---|---|---|---|---|---|
| Task Arithmetic (Ilharco et al., 2023) | 61.1 | 65.9 | 74.0 | 76.2 | 88.0 | 73.9 | 98.4 | 53.0 | 73.8 |
| Ties-Merging (Yadav et al., 2023) | 69.1 | 72.5 | 80.5 | 84.0 | 85.0 | 71.5 | 98.1 | 54.9 | 77.0 |
| **Task-wise AdaMerging** (Ours) | 64.4 | 64.2 | 75.4 | 86.7 | 86.3 | 86.7 | 97.6 | 46.9 | 76.0 |
| **Task-wise AdaMerging++** (Ours) | 67.8 | 70.2 | 79.9 | 89.2 | 87.5 | 79.2 | 98.3 | 51.9 | 78.0 |
| **Layer-wise AdaMerging** (Ours) | 70.2 | 80.7 | 81.6 | 94.8 | 91.6 | 95.8 | 98.5 | 66.2 | 84.9 |
| **Layer-wise AdaMerging++** (Ours) | 71.8 | 80.8 | 84.1 | 94.3 | 91.9 | 94.5 | 98.7 | 69.8 | 85.7 |

**Generalization**. As shown in Tab. 7, we demonstrate the generalization of Layer-wise AdaMerging under the ViT-B/16 architecture. In the two unseen test tasks of EuroSAT and MNIST (their corresponding task vectors are not merged), our AdaMerging improved the average accuracy by 2.3% compared to Task Arithmetic. On two unseen tasks, RESISC45 and SVHN, the average accuracy increased by 1.1%. This shows that AdaMerging has better generalization properties.

Table 7: Generalization results on two unseen tasks when merging ViT-B/16 models on six tasks.

| Method | Seen Tasks | | | | | | | Unseen Tasks | | |
|---|---|---|---|---|---|---|---|---|---|---|
| | SUN397 | Cars | RESISC45 | DTD | SVHN | GTSRB | Avg Acc | EuroSAT | MNIST | Avg Acc |
| Task Arithmetic | 68.1 | 73.0 | 81.6 | 59.1 | 89.1 | 83.8 | 75.8 | 43.9 | 87.5 | 65.7 |
| **AdaMerging** (Ours) | 69.1 | 79.3 | 90.0 | 66.2 | 95.2 | 94.4 | 82.4 | 45.9 | 90.1 | 68.0 |

| Method | SUN397 | Cars | GTSRB | EuroSAT | DTD | MNIST | Average | RESISC45 | SVHN | Average |
|---|---|---|---|---|---|---|---|---|---|---|
| Task Arithmetic | 69.0 | 73.8 | 81.1 | 87.6 | 58.2 | 98.4 | 78.0 | 56.0 | 67.7 | 61.8 |
| **AdaMerging** (Ours) | 72.9 | 81.0 | 97.1 | 96.4 | 66.5 | 99.2 | 85.5 | 52.3 | 75.6 | 63.9 |

**Robustness**. Tab. 8 shows the robustness test of AdaMerging and Task Arithmetic based on ViT-B/16 on seven corruption test datasets. Fig. 5 shows an example of corruption. We can observe that in the test datasets Motion Blur, Impulse Noise, Gaussian Noise, Pixelate, Spatter, Contrast and JPEG Compression where the distribution drifts, the average accuracy of AdaMerging is 9.9%, 8.2%, 7.8%, 6.8%, 12.4%, 9.5% and 9.7% higher than that of Task Arithmetic, respectively. This shows that our AdaMerging is more robust to test data distribution shifts than Task Arithmetic.

## B.2  ANALYSIS EXPERIMENT

**Task Relationship Analysis**. As shown in Fig. 6(a) and (b), we show the correlation between pairs of task vectors in ViT-B/32 and ViT-L/14, respectively. We observe a phenomenon consistent with Task Arithmetic (Ilharco et al., 2023), that is, these task vectors are almost orthogonal to each other.

Table 8: Robustness results when merging ViT-B/16 models on four tasks.

| Method | Clean Test Set | | | | | Corruption Test Set (Motion Blur) | | | | |
| --- | --- | --- | --- | --- | --- | --- | --- | --- | --- | --- |
| | Cars | EuroSAT | RESISC45 | GTSRB | **Avg Acc** | Cars | EuroSAT | RESISC45 | GTSRB | **Avg Acc** |
| Task Arithmetic | 75.3 | 96.3 | 85.3 | 80.5 | 84.3 | 73.5 | 70.9 | 83.9 | 72.2 | 75.1 |
| **AdaMerging** (Ours) | 83.4 | 97.2 | 88.6 | 97.5 | 91.7 | 81.3 | 75.9 | 87.4 | 95.6 | 85.0 |

| Method | Corruption Test Set (Impulse Noise) | | | | | Corruption Test Set (Gaussian Noise) | | | | |
| --- | --- | --- | --- | --- | --- | --- | --- | --- | --- | --- |
| | Cars | EuroSAT | RESISC45 | GTSRB | **Avg Acc** | Cars | EuroSAT | RESISC45 | GTSRB | **Avg Acc** |
| Task Arithmetic | 70.4 | 59.5 | 75.2 | 54.0 | 64.8 | 72.2 | 60.8 | 78.5 | 51.0 | 65.6 |
| **AdaMerging** (Ours) | 77.6 | 42.1 | 81.9 | 90.2 | 73.0 | 79.1 | 58.9 | 81.2 | 74.5 | 73.4 |

| Method | Corruption Test Set (Pixelate) | | | | | Corruption Test Set (Spatter) | | | | |
| --- | --- | --- | --- | --- | --- | --- | --- | --- | --- | --- |
| | Cars | EuroSAT | RESISC45 | GTSRB | **Avg Acc** | Cars | EuroSAT | RESISC45 | GTSRB | **Avg Acc** |
| Task Arithmetic | 03.8 | 38.0 | 24.8 | 71.3 | 34.5 | 72.1 | 58.4 | 79.9 | 60.1 | 67.6 |
| **AdaMerging** (Ours) | 04.1 | 46.4 | 23.6 | 91.3 | 41.3 | 79.3 | 60.9 | 85.8 | 93.7 | 80.0 |

| Method | Corruption Test Set (Contrast) | | | | | Corruption Test Set (JPEG Compression) | | | | |
| --- | --- | --- | --- | --- | --- | --- | --- | --- | --- | --- |
| | Cars | EuroSAT | RESISC45 | GTSRB | **Avg Acc** | Cars | EuroSAT | RESISC45 | GTSRB | **Avg Acc** |
| Task Arithmetic | 73.4 | 62.5 | 81.3 | 76.9 | 73.5 | 75.1 | 73.1 | 84.8 | 64.7 | 74.4 |
| **AdaMerging** (Ours) | 81.4 | 68.1 | 85.8 | 96.8 | 83.0 | 81.9 | 76.0 | 87.3 | 91.0 | 84.1 |

Figure 5: An example of corruption data visualization, in which the corruption image generation method refers to Hendrycks & Dietterich (2019).

In particular, there are very few task vectors with high similarity between them, such as SVHN and MNIST, because they are both handwritten digit recognition tasks. The orthogonality of task vectors provides good initial conditions for model merging, indicating that they have the potential to be combined into a single model, and our results show that this is indeed the case. Further, we merge four groups of task vectors with different correlation degrees, namely (SVHN, MNIST), (SVHN, GTSRB), (SVHN, SUN397), and (SVHN, EuroSAT). The results are shown in Fig. 7. We observe that under task vectors with different degrees of correlation, our AdaMerging technique is always effective because it aims to adaptively learn optimal merging coefficients.

**Impact of the Amount of Available Test Data on Performance.** The AdaMerging proposed in this paper requires an unlabeled test dataset to perform entropy minimization optimization. Having all test data available may be unrealistic in some scenarios. In this section, we verify the performance changes of AdaMerging when different amounts (e.g., 0.1%, 1%, 5%, 100%) of test data are available. As shown in Fig. 8 and Tab. 9, we observed that even when only 0.1% of unlabeled test data are available, our AdaMerging and AdaMerging++ still have a performance improvement of 4.9% and

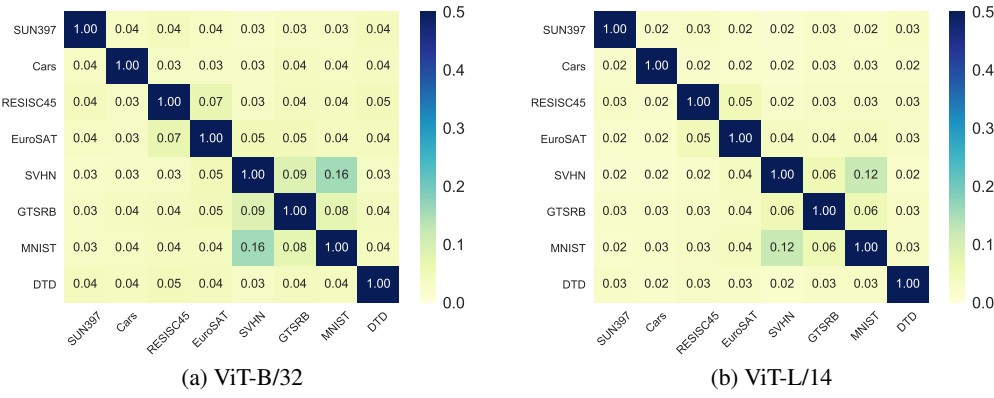

Figure 6: Cosine similarity between task vectors on ViT-B/32 and ViT-L/14.

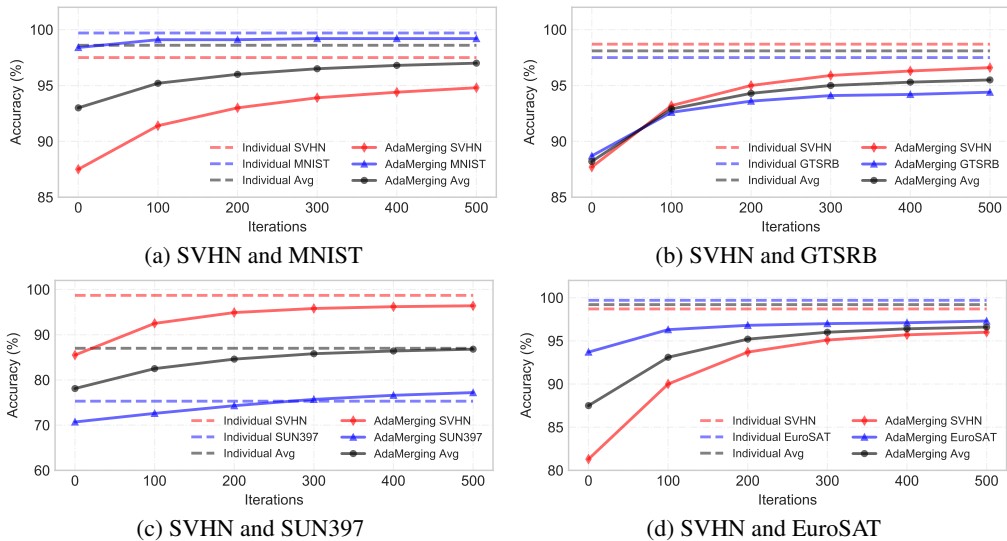

Figure 7: Merging of task vectors with different correlations on the ViT-B/32 model. Note that when iteration=0, it also represents the performance of Task Arithmetic (Ilharco et al., 2023) ($\lambda = 0.3$).

5.5%, respectively, compared to Task Arithmetic and Ties-Merging. In addition, when 5% of the data are available, it can almost achieve a performance comparable to 100% of the data. This shows that our AdaMerging is valuable and can bring significant performance improvements even with a small amount of data.

Table 9: Impact of the amount of available test data on performance when merging ViT-B/32 models.

| Method | Available TestSet | SUN397 | Cars | RESISC45 | EuroSAT | SVHN | GTSRB | MNIST | DTD | Avg Acc |
|---|---|---|---|---|---|---|---|---|---|---|
| Task Arithmetic (Ilharco et al., 2023) | 0.00% | 55.2 | 54.9 | 66.7 | 78.9 | 80.2 | 69.7 | 97.3 | 50.4 | 69.1 |
| **Layer-wise AdaMerging** (Ours) | 0.10% | 62.5 | 59.7 | 71.2 | 69.5 | 89.4 | 84.2 | 98.2 | 57.5 | 74.0 |
| **Layer-wise AdaMerging** (Ours) | 1.00% | 61.9 | 66.3 | 81.8 | 86.0 | 88.6 | 85.8 | 97.4 | 52.5 | 77.5 |
| **Layer-wise AdaMerging** (Ours) | 5.00% | 63.7 | 68.6 | 79.1 | 93.3 | 86.5 | 91.7 | 97.2 | 61.9 | 80.1 |
| **Layer-wise AdaMerging** (Ours) | 100.0% | 64.5 | 68.1 | 79.2 | 93.8 | 87.0 | 91.9 | 97.5 | 59.1 | 80.1 |
| Ties-Merging (Yadav et al., 2023) | 0.00% | 59.8 | 58.6 | 70.7 | 79.7 | 86.2 | 72.1 | 98.3 | 54.2 | 72.4 |
| **Layer-wise AdaMerging++** (Ours) | 0.10% | 70.0 | 66.2 | 74.6 | 79.9 | 89.3 | 83.6 | 98.4 | 61.2 | 77.9 |
| **Layer-wise AdaMerging++** (Ours) | 1.00% | 66.9 | 68.6 | 81.4 | 91.8 | 89.2 | 87.1 | 98.1 | 61.8 | 80.6 |
| **Layer-wise AdaMerging++** (Ours) | 5.00% | 66.4 | 68.4 | 81.5 | 92.9 | 90.0 | 89.0 | 98.2 | 61.5 | 81.0 |
| **Layer-wise AdaMerging++** (Ours) | 100.0% | 66.6 | 68.3 | 82.2 | 94.2 | 89.6 | 89.0 | 98.3 | 60.6 | 81.1 |

**Supervised AdaMerging Analysis**. This paper uses unsupervised entropy minimization as a proxy objective for supervised cross-entropy loss to optimize model merging coefficients. Therefore, AdaMerging trained with supervised cross-entropy loss should be an *upper bound* on our unsupervised AdaMerging. As shown in Fig. 9 and Tab. 10, we observe that the performance of our unsupervised AdaMerging version is very close to that of the supervised AdaMerging version. For example, the Avg Acc of supervised Task-wise AdaMerging is 71.3%, while the Avg Acc of our unsupervised Task-wise AdaMerging is 71.1%. This also further verifies that it is reasonable for us to use entropy minimization as a proxy for cross-entropy loss in merging coefficients learning.

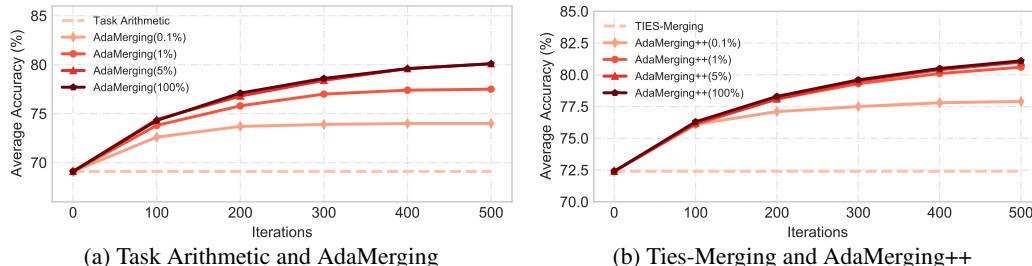

(a) Task Arithmetic and AdaMerging      (b) Ties-Merging and AdaMerging++

Figure 8: Impact of the amount of available test data (e.g., $0.1\%, 1\%, 5\%, 100\%$) on performance when merging ViT-B/32 models.

Table 10: Performance comparison between supervised and unsupervised versions of AdaMerging.

| Method | Label | SUN397 | Cars | RESISC45 | EuroSAT | SVHN | GTSRB | MNIST | DTD | Avg Acc |
|---|---|---|---|---|---|---|---|---|---|---|
| Task Arithmetic (Ilharco et al., 2023) | - | 55.2 | 54.9 | 66.7 | 78.9 | 80.2 | 69.7 | 97.3 | 50.4 | 69.1 |
| **Task-wise AdaMerging** | Supervised | 58.4 | 56.4 | 74.8 | 81.2 | 81.5 | 77.4 | 88.3 | 52.3 | 71.3 |
| **Task-wise AdaMerging** | Unsupervised | 58.0 | 53.2 | 68.8 | 85.7 | 81.1 | 84.4 | 92.4 | 44.8 | 71.1 |
| **Layer-wise AdaMerging** | Supervised | 66.8 | 68.4 | 85.3 | 92.4 | 88.7 | 89.8 | 95.9 | 65.6 | 81.6 |
| **Layer-wise AdaMerging** | Unsupervised | 64.5 | 68.1 | 79.2 | 93.8 | 87.0 | 91.9 | 97.5 | 59.1 | 80.1 |
| Ties-Merging (Yadav et al., 2023) | - | 59.8 | 58.6 | 70.7 | 79.7 | 86.2 | 72.1 | 98.3 | 54.2 | 72.4 |
| **Task-wise AdaMerging++** | Supervised | 61.6 | 59.3 | 77.8 | 80.1 | 84.8 | 79.1 | 91.5 | 55.1 | 73.7 |
| **Task-wise AdaMerging++** | Unsupervised | 60.8 | 56.9 | 73.1 | 83.4 | 87.3 | 82.4 | 95.7 | 50.1 | 73.7 |
| **Layer-wise AdaMerging++** | Supervised | 68.2 | 69.8 | 84.8 | 93.4 | 89.3 | 89.1 | 97.3 | 64.3 | 82.0 |
| **Layer-wise AdaMerging++** | Unsupervised | 66.6 | 68.3 | 82.2 | 94.2 | 89.6 | 89.0 | 98.3 | 60.6 | 81.1 |

**Parameter Cost Analysis**. As shown in Tab. 11, our AdaMerging introduces very few coefficients that need to be updated. The total number of parameters of the eight task vectors is 907,589,640, and our Task-wise AdaMerging only added 8 parameters, and Layer-wise AdaMerging added 1,248 parameters.

**Time Cost Analysis**. As shown in Tab. 12, we show the performance that AdaMerging can achieve under different training costs (based on a single GeForce RTX 3090). We observed that our AdaMerging brought about a 2% performance improvement when it took 7.5 minutes longer than Task Arithmetic. When training for 50 minutes, AdaMerging brought an 8% performance improvement. This shows that AdaMering is very cost-effective and can bring significant performance improvements with only a small amount of training time.

**Merging Coefficients Visualization**. Fig. 11 shows the changes during the iteration process of merging coefficient optimization of each task vector in Task-wise AdaMerging and AdaMerging++, which is shown every ten steps. In addition, Fig. 12 and Fig. 13 show the merging coefficients of eight task vectors learned by Layer-wise AdaMerging and AdaMerging++ on ViT-B/16 respectively. Finally, Fig. 14 and Fig. 15 show the coefficients learned under ViT-L/14. We can clearly observe that in different layers of different task vectors, the learned merging coefficients are different. Finding the merging coefficients of so many layers through grid search is almost impossible.

**Visualization of Spearman's Correlation Coefficient Between Entropy and Loss**. As shown in Fig. 10, we show the correlation changes of unsupervised entropy minimization and supervised cross-entropy loss at different training stages (i.e., the number of iterations are {0, 100, 200, 300, 400, 500} respectively). We observe that in the merging coefficients learning process of AdaMerging, entropy minimization and cross-entropy loss always have a high correlation. Therefore, entropy minimization can be used as a surrogate objective to optimize model merging coefficients.

**Visualization of Correlation between Entropy and Loss**. As shown in Fig. 16, we analyze the correlation between the entropy and the model's prediction loss for eight tasks (or datasets) on the initial merged model. As described in Sec. 3.2.2, in each dataset, we sort the entropy on the test samples from small to large into eleven groups and observe the average loss of sample prediction within each group. We observe that groups with smaller entropy generally have smaller average losses. Therefore, it is reasonable to take Shannon entropy minimization as an unsupervised optimization surrogate objective for loss (e.g., cross-entropy) minimization.

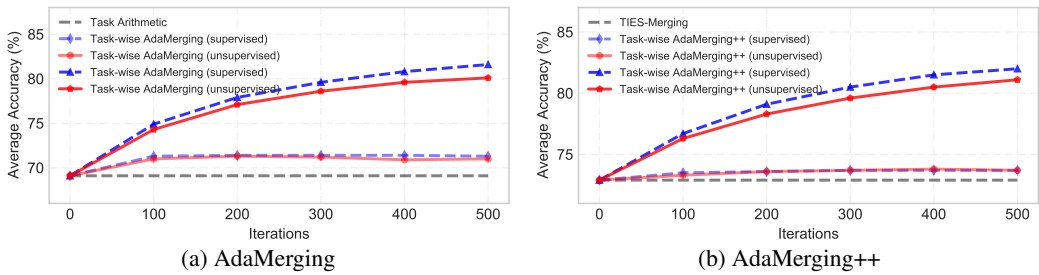

Figure 9: Supervised and Unsupervised AdaMerging/AdaMerging++ merging ViT-B/32 models.

Table 11: Parameter cost of model merging for AdaMering on ViT-B/32.

| Method | Task Arithmetic | Ties-Merging | Task-wise AdaMerging | Layer-wise AdaMerging |
|---|---|---|---|---|
| Total number of model merging parameters (all task vectors) | 907,589,640 | 907,589,640 | 907,589,640 | 907,589,640 |
| Total number of trainable model merging coefficients | - | - | 8 | 1,248 |

Table 12: Time cost of model merging for AdaMering on ViT-B/32.

| Training Time | Base | +7.5 min | +12.5 min | +25 min | +50 min | +100 min | +125 min |
|---|---|---|---|---|---|---|---|
| Avg Acc of Layer-wise AdaMerging | 69.1 | 71.1 | 72.1 | 74.5 | 77.1 | 79.7 | 80.1 |
| Avg Acc of Layer-wise AdaMerging++ | 72.4 | 74.1 | 74.8 | 76.3 | 78.3 | 80.5 | 81.1 |

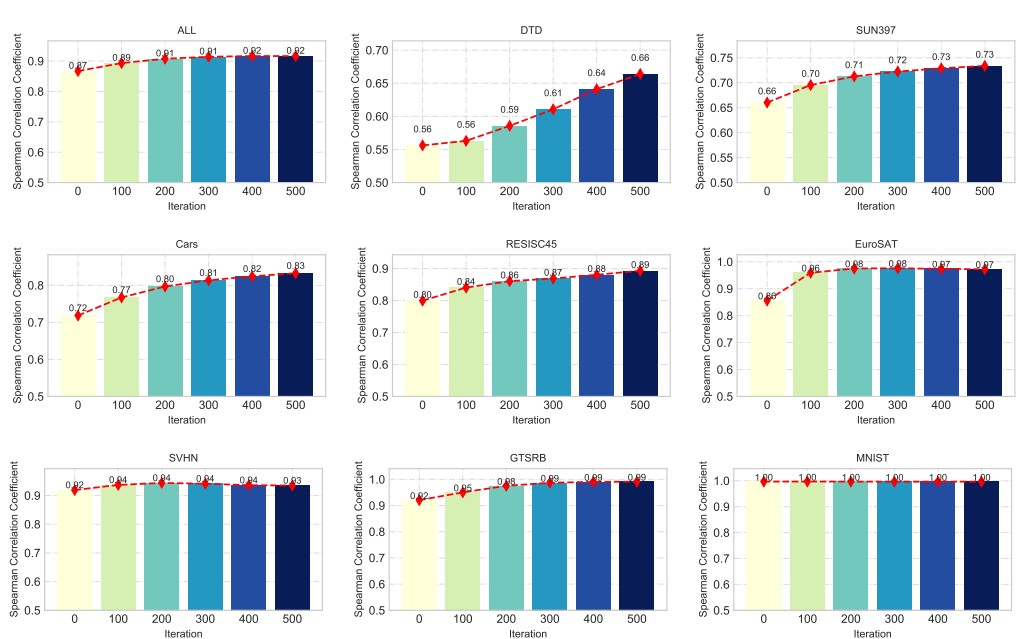

Figure 10: Spearman correlation coefficient between *entropy* $H(\hat{Y})$ and *avareage loss* $L(Y, \hat{Y})$ on eight tasks (or datasets) at different stages of training(e.g., Iteration=$\{0, 100, 200, 300, 400, 500\}$), and we observed a high positive correlation.

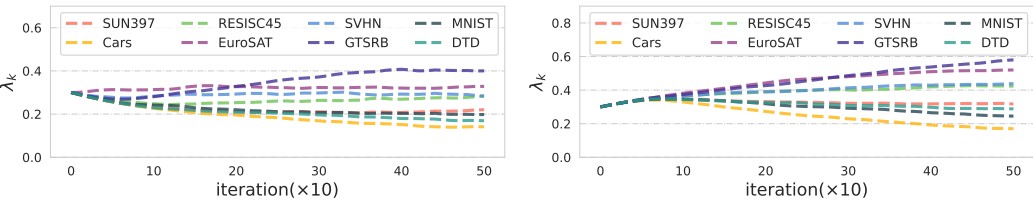

Figure 11: Model merging coefficients $\{\lambda_k\}_{k=1}^K$ change with respect to training steps on ViT-B/32: (a) Task-wise AdaMerging; (b) Task-wise AdaMerging++. Each line represents the change process of the coefficient $\lambda_k$ of a task vector $T_k$ ($k \in \{1, 2, \ldots, K\}$).

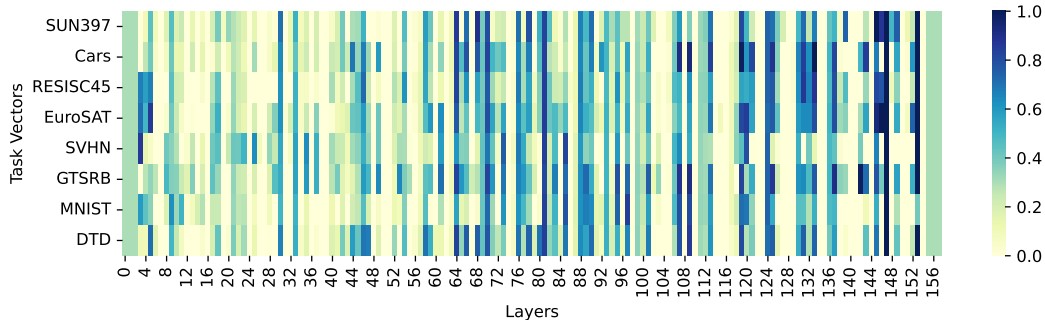

Figure 12: Learned model merging coefficients of **Layer-wise AdaMerging on ViT-B/16**. The $k$-th row represents the $k$-th task vector, the $l$-th column represents the $l$-th layer, and the intersection point represents the coefficient $\lambda_k^l$.

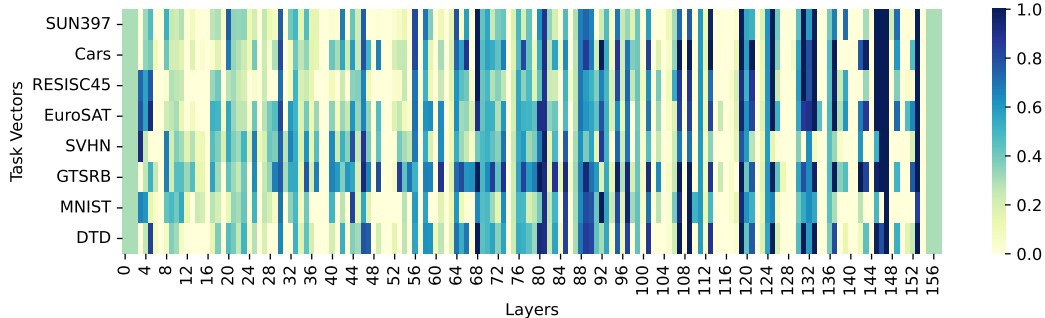

Figure 13: Learned model merging coefficients of **Layer-wise AdaMerging++ on ViT-B/16**. The $k$-th row represents the $k$-th task vector, the $l$-th column represents the $l$-th layer, and the intersection point represents the coefficient $\lambda_k^l$.

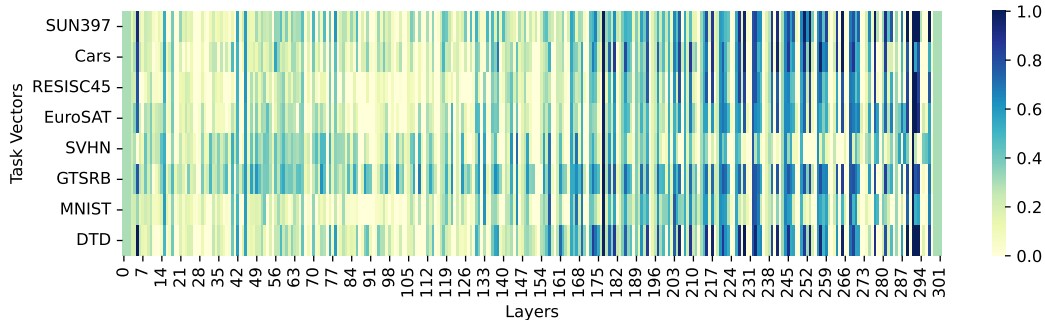

Figure 14: Learned model merging coefficients of **Layer-wise AdaMerging on ViT-L/14**. The $k$-th row represents the $k$-th task vector, the $l$-th column represents the $l$-th layer, and the intersection point represents the coefficient $\lambda_k^l$.

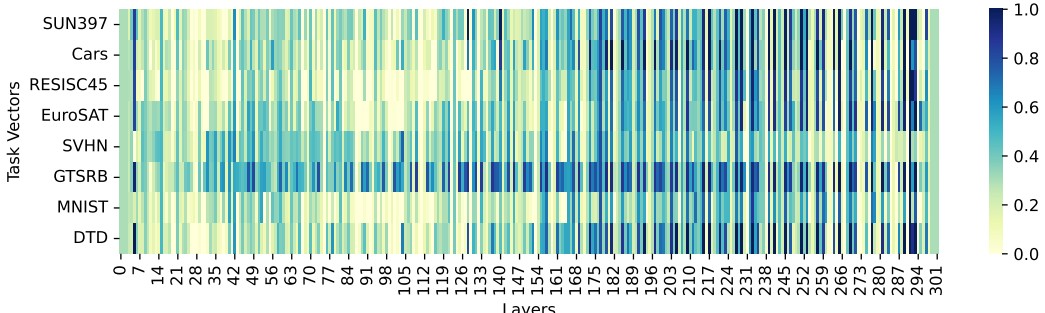

Figure 15: Learned model merging coefficients of **Layer-wise AdaMerging++ on ViT-L/14**. The $k$-th row represents the $k$-th task vector, the $l$-th column represents the $l$-th layer, and the intersection point represents the coefficient $\lambda_k^l$.

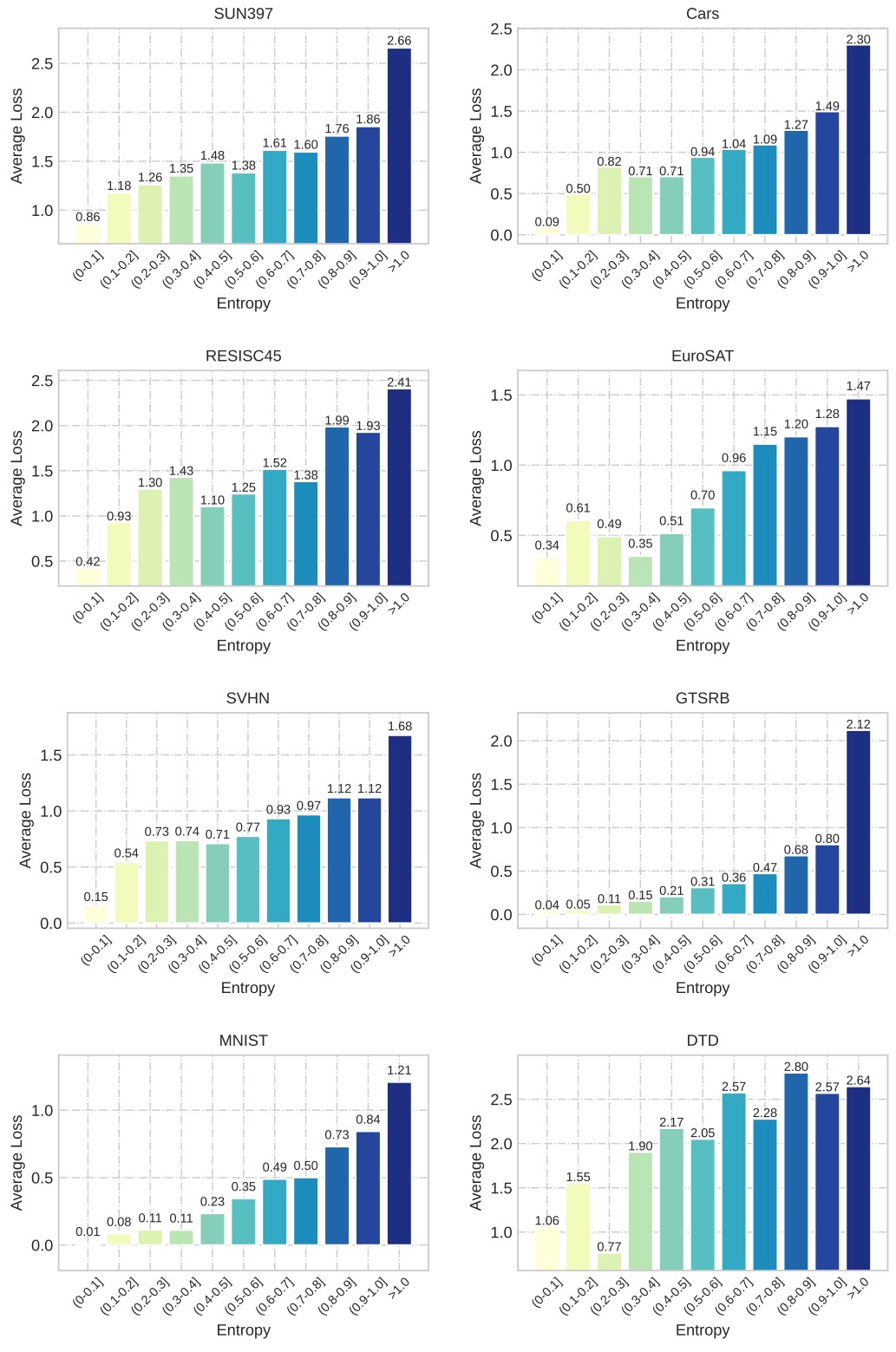

Figure 16: Correlation analysis of *entropy* and *average loss* on eight tasks (or datasets). We can observe that there is a **high positive correlation** between entropy and prediction loss on each dataset.

