# OpenReview forum: "AdaMerging: Adaptive Model Merging for Multi-Task Learning"
_ICLR.cc/2024/Conference — ICLR 2024 poster_

### Official Review · Reviewer_zdU4 · 2023-10-21

**Soundness:** 2 fair
**Presentation:** 3 good
**Contribution:** 1 poor
**Rating:** 6
**Confidence:** 4

**Summary:**

This paper presents a model merging method in the context of multi-task learning. The key idea is to take a pre-trained model and fine-tune it separately for each task using task-specific data, resulting in task-specific models. Then the paper introduces a novel method for automatically merging the task-specific parameters without the need for retraining. In essence, this work builds upon the foundations of Task Arithmetic [1] and TIES-MERGING [2] but enhances the process by incorporating adaptive task weights.

[1] Ilharco, Gabriel, et al. "Editing models with task arithmetic." ICLR2023.
[2] Yadav, Prateek, et al. "Resolving Interference When Merging Models." NIPS2023.

**Strengths:**

- The paper is easy to follow.
- The idea is easy to catch up with.

**Weaknesses:**

- The novelty remains a question for me.
- The practical value of this method is not well supported.
Please refer to the questions part.

**Questions:**

- I'm not sure the paper has sufficient novelty to be published in the top-tier conference since the proposed method only goes one step further from Task Arithmetic [1] and TIES-MERGING [2] by incorporating trainable weights for task vectors.The concept seems thin to support an entire paper, with only one page (page 6) dedicated to the novel part. Authors should consider diving deeper into this direction. For example, exploring the underlying reasons for the weight relationships between different tasks and their potential correlation with task relationships could enhance the paper's depth. Additionally, the learned weights could be utilized to guide the training of multi-task models, as seen in Auto-lambda [3].

- Is it really necessary to conduct experiments to show the relationship between Shannon entropy and cross entropy? Actually from the information theory, the two concepts are almost the same thing or we can say cross entropy is derived from Shannon entropy. It's kind of trivial or even unnecessary to do experiments in Figure 3. Besides, the usage of Shannon entropy to train the adaptive weights also limits the method can only be used for classification tasks.

- It's better to show the performance of the pre-trained model on each task as well in Tables 1 and 2.

- Limited application of this kind of work. From Tables 1 and 2, we can clearly see that traditional MTL or we say all-shared MTL can achieve a very high accuracy, not to say SOTA MTL methods like AdaShare [4] and AutoMTL[5]. In practice, machine learning engineers might prefer these alternatives due to their superior performance. Besides, for the model merging direction, it's weird to assume that although we may not be able to get the train data for each task, we can still get the pre-trained weights of the model. Most importantly, those task-specific models even need to be trained from the same pre-trained weights.

[1] Ilharco, Gabriel, et al. "Editing models with task arithmetic." ICLR2023.
[2] Yadav, Prateek, et al. "Resolving Interference When Merging Models." NIPS2023.
[3] Liu, Shikun, et al. "Auto-lambda: Disentangling dynamic task relationships." TMLR2022.
[4] Sun, Ximeng, et al. "Adashare: Learning what to share for efficient deep multi-task learning." NIPS2020.
[5] Zhang, Lijun, Xiao Liu, and Hui Guan. "Automtl: A programming framework for automating efficient multi-task learning." NIPS2022.

---

> ### Author Response · Authors · 2023-11-21
>
> Thank you very much for your review and affirmation of our work. We'll answer your questions one by one below.
>
> ## 1. About the question of "The novelty of this paper":
>
> (1) The research direction of this paper is an important direction, which is to merge models trained independently on multiple tasks into one to perform MTL. It can reduce the effort of collecting original training data and avoid the cost of model retraining. However, there is still a significant performance gap between SOTA model merging schemes and traditional MTL, and we observe that the model merging coefficient has a very large impact on performance. In this paper, we first propose AdaMerging to automatically learn Task-wise or Layer-wise merging coefficients. Further, we verify through extensive experiments that unsupervised entropy minimization can be used as a proxy objective for supervised cross-entropy loss to optimize model merging coefficients. Our approach, although seemingly simple, is highly effective. For example, compared with two SOTA task vector-based methods, Task Arithmetic (ICLR'2023) and Ties-Merging (NeurIPS'2023), our AdaMerging and AdaMerging++ bring up to 11% and 8.7% performance improvement, respectively. Even if only 0.1% of the test data is available, our method can bring about a 5% performance improvement. Our method also exhibits stronger generalization and robustness.
>
> | Method | Avg ACC (\%) | | Method | Avg ACC (\%) |
> | :------ | :------: |   :------: |   :------ | :------: |
> | Task Arithmetic (0%)  | 69.1 |    |  Ties-Merging (0%)  |  72.4 |
> | Our AdaMerging (0.1%)   | 74.0 |  |  Our AdaMerging++ (0.1%) | 77.9  |
> | Our AdaMerging (1%)   | 77.5 |  |  Our AdaMerging++ (1%) | 80.6  |
> | Our AdaMerging (5%)   | 80.1 |  |  Our AdaMerging++ (5%) |   81.0 |
> | Our AdaMerging (100%) | 80.1 |  |  Our AdaMerging++ (100%) |  81.1  |
>
> (2) The tasks and data of our experiments strictly follow the settings of Task Arithmetic[1] and Ties-Merging[2]. We have added similarity calculations for task vectors in the modified version (as shown in Figure 6 in the appendix). We have observed that a small number of task vectors have a certain degree of correlation (such as SVHN and MNIST, which are both digit recognition tasks), and most task vectors have low correlation. That is, these task vectors are almost orthogonal (this is consistent with Task Arithmetic). We did not observe a direct correlation between the 'similarity of the task vectors' and the 'merging coefficients'. We further added task vector merging experiments with different correlation degrees, as shown in Figure 7 in the appendix. We observed that our AdaMerging is effective under various correlation degrees.
>
> (3) The suggestion that "weights learned from task vectors can be used to guide MTL training" may not be suitable for the setting of this paper. Model merging represents only the trained model for each task, without the original training data. Therefore, model merging technology is unsuitable for scenarios where a weighted MTL can be trained directly from scratch. Model merging is mainly used for foundation models or large language models, which are very expensive to train from scratch using raw data.
>
> [1] Ilharco, Gabriel, et al. "Editing models with task arithmetic." ICLR2023.
>
> [2] Yadav, Prateek, et al. "Resolving Interference When Merging Models." NIPS2023.
>
> ## 2. About the question of "The relationship between Shannon entropy and cross entropy":
>
> Indeed, there is a close relationship between Shannon entropy and cross-entropy in information theory. However, they still have obvious differences in merging coefficients training. Entropy minimization does not require labels, while cross-entropy loss requires labeled data. Experimental confirmation of this relationship can provide readers with a more intuitive understanding that entropy minimization can serve as an effective proxy for cross-entropy loss when merging models. In the modified version (Figure 10 in the appendix), our method further increases the correlation between entropy minimization and cross-entropy loss minimization across different training stages.
>
> In addition, although entropy minimization can only be used for classification tasks, classification tasks are very common in various applications in machine learning, and they cover many practical problems, including but not limited to image recognition, natural language processing, bioinformatics, and other fields.

---

> ### Author Response · Authors · 2023-11-21
>
> ## 3. About the question of "The performance of the pre-trained model":
>
> As shown in the below tables, we added the pre-trained model to predict the performance of each task. It can be observed that there is a big gap in performance between pre-trained and Task Arithmetic. Our AdaMerging significantly outperforms them.
>
> | Method (merging ViT-B/32)  | SUN397 | Cars | RESISC45 | EuroSAT | SVHN | GTSRB | MNIST | DTD | Avg |
> | :------ | :------: |  :------: |:------: |  :------: |:------: |  :------: |:------: |  :------: | :------: |
> | Pre-trained   | 62.3 | 59.7 | 60.7  | 45.5  | 31.4  | 32.6  | 48.5  | 43.8  | 48.0  |
> | Task Arithmetic   |  55.2 | 54.9  | 66.7  | 78.9 |  80.2 |  69.7 |  97.3 |  50.4  | 69.1 |
> | AdaMerging(Ours) | 64.5 | 68.1 | 79.2 | 93.8 | 87.0 | 91.9 | 97.5 | 59.1 | 80.1|
>
> | Method (merging ViT-L/14)  | SUN397 | Cars | RESISC45 | EuroSAT | SVHN | GTSRB | MNIST | DTD | Avg |
> | :------ | :------: |  :------: |:------: |  :------: |:------: |  :------: |:------: |  :------: | :------: |
> | Pre-trained   | 66.8  | 77.7  | 71.0  |  59.9 | 58.4  | 50.5  | 76.3  | 55.3  | 64.5  |
> | Task Arithmetic   | 73.9 | 82.1 | 86.6 | 94.1 | 87.9 | 86.7 | 98.9 | 65.6 | 84.5|
> | AdaMerging(Ours) | 79.0 | 90.3 | 90.8 | 96.2 | 93.4 | 98.0 | 99.0 | 79.9 | 90.8|
>
> ## 4. About the question of "The limited application of this kind of work":
>
> At this stage, model merging has a certain performance gap with traditional MTL, but it is a potential and popular research direction [1-8]. The traditional MTL method requires the original data of all tasks to be collected together in advance, and then iteratively updates the MTL model, which has the risk of data privacy leakage. Especially in the era of large models, this method of training MTL models from scratch is inefficient. Model merging does not require original task training data, but only requires models trained independently for each task to complete MTL. Reviewer eRin also noted that "developing a single versatile model from diverse off-the-shelf fine-tuned models is important to LLM (or Foundation model) communities".
>
> In addition, regarding the issue of relying on the same pre-trained model. In the era of large models, the backbone of most research work is highly homogeneous and is often based on the same pretrain model. Ties-Merging [1] points out that "these benefits have resulted in the release of *thousands* of fine-tuned checkpoints derived from popular PTMs such as ViT for vision and T5 for language." In addition, learning from models [2] rather than raw data is a critical research direction.
>
> [1] Yadav, Prateek, et al. "Resolving Interference When Merging Models." NIPS 2023.
>
> [2] Zheng, Hongling, et al. "Learn From Model Beyond Fine-Tuning: A Survey." arXiv preprint arXiv:2310.08184 (2023).
>
> [3] Ilharco, Gabriel, et al. "Editing models with task arithmetic." ICLR2023.
>
> [4] Zhang, Jinghan, et al. "Composing parameter-efficient modules with arithmetic operations." arXiv preprint arXiv:2306.14870 (2023).
>
> [5] Li, Weishi, et al. "Deep model fusion: A survey." arXiv preprint arXiv:2309.15698 (2023).
>
> [6] Wang, Song, et al. "Knowledge Editing for Large Language Models: A Survey." arXiv preprint arXiv:2310.16218 (2023).
>
> [7] Chronopoulou, Alexandra, et al. "Language and Task Arithmetic with Parameter-Efficient Layers for Zero-Shot Summarization." arXiv preprint arXiv:2311.09344 (2023).
>
> [8] Ortiz-Jimenez, Guillermo, Alessandro Favero, and Pascal Frossard. "Task Arithmetic in the Tangent Space: Improved Editing of Pre-Trained Models." arXiv preprint arXiv:2305.12827 (2023).

---

> > ### Comment · Reviewer_zdU4 · 2023-11-21
> > **Response to the author**
> >
> > Thank you for your clarification! Most of my concerns are solved. I'll raise my score.

---

> > > ### Author Response · Authors · 2023-11-21
> > >
> > > Thank you for your suggestions to improve our paper and your final support for our work.

---

### Official Review · Reviewer_eRin · 2023-10-31

**Soundness:** 3 good
**Presentation:** 3 good
**Contribution:** 3 good
**Rating:** 6
**Confidence:** 5

**Summary:**

Merging multiple fine-tuned models without a retraining process along with initial training data has been shown to be feasible, i.e., two existing works, Ties-Merging and Task Arithmetic. However, directly adding models may fail due to potential conflicts and intricate correlations. This paper proposed a new approach to automatically learn the merging weights by minimizing testing-time entropy on unlabeled samples in an unsupervised manner. Instead of the need for initial training data, this paper showed that testing-time entropy can serve as an approximated objective compared to traditional supervised loss. The authors proposed two main ways to learn the merging weights. One is task-wise merging, which learns the coefficient for each task. The other is a more fine-grained version, layer-wise merging, which not only learns the coefficient for each task but also for each layer. In their experiments, they included eight tasks to validate their approach with ViT models. Results have shown improvements in performance (for classification accuracy), generalization capabilities (to unseen tasks), and robustness (to test data distribution shifts) compared to the other two SOTA methods.

**Strengths:**

* Originality: Merging multiple fine-tuned models has been shown feasible, but this paper proposed and proved that using testing-time entropy as an objective to learn merging weights is effective and can be automatic. They also suggested that learning weights across different layers is crucial to the success of merging. These show the strength in originality.
* Quality: The results of the experiments are solid and promising. They closed the performance gap between conventional MTL and task arithmetic-based methods.
* Clarity: The presentation of this paper is clear.
* Significance: Developing a single versatile model from ***diverse off-the-shelf fine-tuned models*** is important to LLM communities. Methods proposed by this work can reduce the efforts of collecting initial training data and the need for retraining for merging models. Besides, to merge fine-tuned models without tuning merging weights by grid-search, their automatic method to learn weights via test-time entropy is important to develop the means of model fusion. At last, the results in the paper improved two existing SOTA methods and showed method's strength in several dimensions.

**Weaknesses:**

* Though we don’t need to train model again via original training data, we still need to access a certain amount of testing data for testing-time-entropy minimization. How the (minimum) amount of testing data can affect the quality of merging weights ($\lambda$ in the paper) is encouraged to study and present in the paper.
* In additional to the amount of testing data, the burden/computational needs/computational time to learn $\lambda$ to converge via unlabeled testing samples is missing and lack of comparison to the other two SOTA methods. This study should be included and enhance the soundness of the proposed method.
* I didn't see any restriction or regularization on $\lambda$, especially in the optimization objective in Sec. 3.2.2. Does $\lambda$ always need to be $\sum_{i=k}^K \lambda_i = 1$?

**Questions:**

* The task relationships across 8 tasks included in the paper can be mentioned. I am curious about the performance changes when we have unrelated tasks and high-correlated tasks.
* In some cases, can the part of the merging weights be negative terms?
* Does the $\lambda$ correlate to the performance gain in Table 1 and Table 2? Is there any relationship between weights and performance gain/loss?
* (minor comment): It will be nice to move Fig 2 earlier for a better understanding.

---

> ### Author Response · Authors · 2023-11-21
>
> Thank you very much for your review and affirmation of our work. We'll answer your questions one by one below.
>
> ## 1. About the question of "How does the (minimum) amount of test data affect performance":
>
> Indeed, using all test data for model pooling coefficient training may not be feasible in some scenarios. Based on your suggestions, we experiment with the performance of AdaMerging when different numbers (such as 0.1%, 1%, 5%, 100%) of test sets are available, as shown in the table below. We observe that even when only 0.1% of the test data is available, AdaMerging and AdaMerging++ have about 5% performance improvement compared to Task Arithmetic and Ties-Merging. When 5% of the test data is available, AdaMerging and AdaMerging++ can almost reach the performance when all test data is available. This significantly improves the applicability of our work.
> | Method | Avg ACC (\%) | | Method | Avg ACC (\%) |
> | :------ | :------: |   :------: |   :------ | :------: |
> | Task Arithmetic (0%)  | 69.1 |    |  Ties-Merging (0%)  |  72.4 |
> | AdaMerging (0.1%)   | 74.0 |  |  AdaMerging++ (0.1%) | 77.9  |
> | AdaMerging (1%)   | 77.5 |  |  AdaMerging++ (1%) | 80.6  |
> | AdaMerging (5%)   | 80.1 |  |  AdaMerging++ (5%) |   81.0 |
> | AdaMerging (100%) | 80.1 |  |  AdaMerging++ (100%) |  81.1  |
>
> ## 2. About the question of "The burden/computational needs/computational time":
>
> In the table below, we show the time cost of AdaMerging compared to Task Arithmetic. The additional time cost of our complete AdaMerging is about two hours. However, our method also has a 2% improvement when the additional adjustment cost is only 7.5 minutes (based on a single GeForce RTX 3090) more than Task Arithmetic. When the extra cost is 50 minutes, our method has an 8% performance improvement.
> | Method | Training Time Cost (min) | Average Accuracy (%) |
> | :------ | :------: | :------: |
> | Task Arithmetic    | 0.0 | 69.1 |
> | AdaMerging         | 7.5 | 71.1 |
> | AdaMerging         |  12.5   | 72.1 |
> | AdaMerging         |  25  | 74.5 |
> | AdaMerging         |  50  | 77.1 |
> | AdaMerging         |  100  | 79.7 |
> | AdaMerging         |  125  | 80.1 |
>
> In addition, we also show the parameter cost required to train our method. On ViT-B/32, there are only 8 new coefficients for task-level AdaMerging and 1248 new coefficients for hierarchical AdaMerging. This is trivial compared to the 907,589,640 parameters in the task vectors.
>
> | Method | Total number of model merging parameters |  Total number of trainable model merging coefficients |
> | :------ | :------: | :------: |
> | Task Arithmetic / Ties-Merging       |  907,589,640   | 0
> | Task-wise AdaMerging       |  907,589,640  | 8 |
> | Layer-wise AdaMerging     |  907,589,640  | 1,248 |
>
> ## 3. About the question of "Restriction or regularization on $\lambda$":
>
> We do not require $\sum_i \lambda_i =1$, as shown in the visualization of the coefficients in the paper and the merging coefficients learned by Task-wise AdaMerging in the table below.
> | Task Weight | SUN397 | Cars | RESISC45 | EuroSAT | SVHN | GTSRB | MNIST | DTD |
> | :------ | :------: |  :------: |:------: |  :------: |:------: |  :------: |:------: |  :------: |
> | AdaMerging  | 0.2202 | 0.1413 | 0.2826 | 0.3284 | 0.2841 | 0.4003 | 0.1978 | 0.1692 |
> | AdaMerging++| 0.3171 | 0.1698 | 0.4235 | 0.5198 | 0.4386 | 0.5803 | 0.2452 | 0.2885 |

---

> ### Author Response · Authors · 2023-11-21
>
> ## 4. About the question of "Task relationships across 8 tasks":
>
> We have added similarity calculations for task vectors in the modified version (as shown in Figure 6 in the Appendix). As shown in the table, we have observed that a small number of task vectors have a certain degree of correlation (such as SVHN and MNIST, which are both digit recognition tasks). Most of the task vectors have low correlation, which means that these task vectors are almost orthogonal (this is consistent with Task Arithmetic). In addition, in the modified version, we further added task vector merging experiments with different degrees of correlation (such as merging SVHN with MNIST, GTSRB, SUN397 and EuroSAT, respectively). As shown in Figure 7, we observe that our AdaMerging is effective when performing model merging at various correlation levels.
>
> |  | SUN397 | Cars | RESISC45 | EuroSAT | SVHN | GTSRB | MNIST | DTD |
> | :------ | :------: |  :------: |:------: |  :------: |:------: |  :------: |:------: |  :------: |
> | **Cars**     | 1.00  | 0.03  | 0.04  | 0.03  | 0.02  | 0.03  | 0.03 | 0.04 |
> | **Cars**     | 0.03  | 1.00  | 0.03  | 0.03  | 0.02  | 0.03  | 0.03 | 0.03 |
> | **RESISC45** | 0.04  | 0.03  | 1.00  | 0.07  | 0.03  | 0.04  | 0.03 | 0.04 |
> | **EuroSAT**  | 0.03  | 0.03  | 0.07  | 1.00  | 0.05  | 0.04  | 0.04 | 0.04 |
> | **SVHN**     | 0.02  | 0.02  | 0.03  | 0.05  | 1.00  | 0.08  | 0.15 | 0.03 |
> | **GTSRB**    | 0.03  | 0.03  | 0.04  | 0.04  | 0.08  | 1.00  | 0.07 | 0.04 |
> | **MNIST**    | 0.03  | 0.03  | 0.03  | 0.04  | 0.15  | 0.07  | 1.00 | 0.03 |
> | **DTD**      | 0.04  | 0.03  | 0.04  | 0.04  | 0.03  | 0.04  | 0.03 | 1.00 |
>
>
> ## 5. About the question of "The negative merging weights":
>
> We do not limit the value of merging coefficients $\lambda$. However, as shown in the table below, none of the coefficients learned in our experiments are negative. In extreme cases, it may be negative, but tasks with significant negative correlations generally do not merge together. Negative task vectors indicate a severe task conflict, which also means that these task vectors are unsuitable to coexist in a multi-task model.
>
> | Task Weight | SUN397 | Cars | RESISC45 | EuroSAT | SVHN | GTSRB | MNIST | DTD |
> | :------ | :------: |  :------: |:------: |  :------: |:------: |  :------: |:------: |  :------: |
> | AdaMerging  | 0.2202 | 0.1413 | 0.2826 | 0.3284 | 0.2841 | 0.4003 | 0.1978 | 0.1692 |
> | AdaMerging++| 0.3171 | 0.1698 | 0.4235 | 0.5198 | 0.4386 | 0.5803 | 0.2452 | 0.2885 |
>
> ## 6. About the question of "The relationship between $\lambda$ and performance":
>
> Yes, $\lambda$ significantly impacts the accuracy of model mergin. For example, in Figure 1, different $\lambda$ are manually set. An inappropriate $\lambda$ will cause the performance of model merging to be unacceptably low. In Table 1 and Table 2, our AdaMerging is adaptively learned $\lambda$, and the coefficients learned by Layer-wise AdaMerging and AdaMerging++ are shown in Figure 5 of the paper. The coefficients learned by Task-wise AdaMerging and AdaMerging++ are shown in the following table.
> | Task Weight | SUN397 | Cars | RESISC45 | EuroSAT | SVHN | GTSRB | MNIST | DTD |
> | :------ | :------: |  :------: |:------: |  :------: |:------: |  :------: |:------: |  :------: |
> | AdaMerging  | 0.2202 | 0.1413 | 0.2826 | 0.3284 | 0.2841 | 0.4003 | 0.1978 | 0.1692 |
> | AdaMerging++| 0.3171 | 0.1698 | 0.4235 | 0.5198 | 0.4386 | 0.5803 | 0.2452 | 0.2885 |
>
> ## 7. About the question of "Move Figure 2 for ease of understanding":
>
> We have moved Figure 2 to Introduction to facilitate readers' understanding.

---

> > ### Comment · Reviewer_eRin · 2023-11-21
> >
> > Thank you for addressing my questions and concerns. It would be beneficial to include information about the usage of testing data and the time cost in the paper.
> >
> > I have one more question. Given that coefficients can be negative and negatively impact the performance, and considering that the task relationship is crucial to the success of this merging framework, what suggestions or explorations can be made regarding the selection of tasks to merge at the beginning? Thanks for your time and response!

---

> ### Author Response · Authors · 2023-11-22
>
> Dear reviewer, thank you for your effort and response!
>
> In our AdaMerging, the optimization goal of the merging coefficients is to reduce entropy on the test data of all tasks (we verified that it is an effective proxy objective for the supervised cross-entropy loss). Therefore, the model merging coefficients will be reasonably optimized, that is, in a direction that is conducive to improving the final performance.
>
> In addition, we have the following suggestions for task selection. Before merging multiple pre-trained models, we can first check the similarity of any two task vectors. It only involves the calculation of the inner product of the vectors, which is very cheap. If there is no negative correlation between task vectors, then we can merge all tasks into one MTL model using Task Arithmetic/Ties-Merging/AdaMerging. If there are some negative correlations between task vectors, it means that they are not suitable to coexist in the same MTL model. We can group tasks, for example, first group tasks that do not conflict with each other (that is, the similarity is positive) into one group, other tasks into one group (or further divide into subgroups), and then merge the task vectors within each group to obtain multiple MTL models.
> Furthermore, we can group task vectors according to layer levels and form branch structures only on layers with negative correlations. This method can also further reduce the number of parameters of the model after grouping and merging.

---

> > ### Comment · Reviewer_eRin · 2023-11-22
> >
> > Thanks for the response. I have no more question to ask. I will raise my score and support this paper to be accepted.

---

> > > ### Author Response · Authors · 2023-11-22
> > >
> > > Dear reviewer, thank you for your valuable comments and support for our work.

---

### Official Review · Reviewer_3WDA · 2023-10-31

**Soundness:** 2 fair
**Presentation:** 4 excellent
**Contribution:** 2 fair
**Rating:** 6
**Confidence:** 3

**Summary:**

This paper tackles the problem of **Multi-task learning** in the context of foundation models: While the most common MTL paradigm is to train a single model on multiple tasks jointly, the paper investigates a different direction of merging single-task networks to form a single unified network. Like standard MTL, this approach can be affected by negative interference between tasks, in that naively merging model weights of conflicting tasks leads to poor performance.
Specifically, the proposed method builds on task arithmetic: While previous works considers simple uniform averaging of task vectors, this paper proposes `AdaMerging` which automatically learns how to weigh each model when merging the tasks, as well as a per-task *and* per-layer variant. The primary goal is to improve the model performance, as task arithmetic approaches still perform worse than standard MTL training. Finally, a last variant `AdaMerging++` which further integrates some ideas from `TiesMerging` (e.g. by removing redundant parameters and sign conflicts in the task vectors before merging them).

In practice, we may not have access to the original single task model training data, but only to a set of potentially unlabeled multi-task data. A key insight of the paper is that the entropy of the MTL model predictions is often correlated with the actual loss on the corresponding samples. Consequently, the proposed method directly optimizes the weights of the task vectors $\lambda$, while minimizing the entropy of the current MTL model's prediction. The proposed `AdaMerging` is evaluated on ViT-backbones from CLIP models, on a suite of computer vision tasks, and compared to task arithmetic methods as well as traditional MTL optimization.

**Strengths:**

- **Good writing**: The paper is well written and easy to follow, also with good illustrative figures.

- **Interesting research direction**: Task vector arithmetic for foundation model is a novel and interesting take on multi-task learning. Extending it to learning the task vector weights seems like a natural and meaningful direction, and very much in-line with automatic loss/gradient weighing scheme in standard multi-task optimization methods.

- **Good set of ablation experiments**: The results on generalization (unseen tasks and corrupted data), as well as the qualitative visualization of the learned task vectors weights are very insightful.

**Weaknesses:**

- The conclusion of **Section 3.2.2** seems a bit strong to me from the conducted experiment: the analysis shows that the entropy and loss of a trained MTL model are nicely correlated, but it does not necessarily mean that they yield equally good directions during training: Doing the same analysis at different timesteps during the MTL model training could show whether and how this correlation holds during training.

- **Discrepancy in supervision** : If I understood correctly, the single-task and MTL baselines use standard supervised training schemes; Task merging baselines (e.g. task arithmetic) are data-free methods and only require the task vectors; finally`AdaMerging` learns task arithmetic weights on unlabelled test samples ($B_k$ on page 6). While these are unsupervised, seeing test samples seems unfair compared to other baselines, especially the task arithmetic ones; it also may not be realistic to have access to a whole (unlabelled) test dataset at once (as opposed to e.g. a few-shot setting)

- The experimental evaluation only considers **ViT-based backbones on small/medium computer vision benchmarks**. This is a bit different from the introduction, which focuses on readily available pretrained foundation models for which the original training data may be unknown. Furthermore, this also raises the question of MTL baselines: there is a very large literature on automatically weighing tasks losses/gradients for computer vision tasks, which may be stronger baselines than `traditional MTL`, and increase the performance gap even more (e.g. GradNorm, PCGrad, GradDrop...etc)


**Overal summary**: To summarize, my main concerns are mainly *(i)* the fairness of chosen baselines (in terms of supervision in the case of task arithmetic schemes, and in terms of optimization strategies for traditional MTL) and to a lesser degree *(ii)* the strength of the conclusions derived from the analysis in **Section 3.2.2**.

**Post rebuttal summary** I'm increasing my rating from 5 to 6 as the authors have addressed my main concerns, in particular about the additional requirement of test data; and I think/hope a more in-depth discussion of the trade-offs of the newly introduced experimental setting compared to the ones of traditional task merging and traditional MTL would make the contribution even stronger.

**Questions:**

- Does the **traditional MTL** baseline use some form of task weighing ? Intuitively I would expect that a "good" set of task vector weights might also be useful for reweighing/rebalancing the different tasks losses/data in traditional MTL training; but it would also be an interesting insight if that is not the case

- Do you have insights on how a **supervised variant of `AdaMerging`** would perform ? It would be interesting to understand how much of the current gap with traditional MTL is due to the different supervision assumption, or due the task arithmetic process itself, versus directly finetuning the model weights  on MTL data.

- Minor note:
  *  in related work: *Task Arthmetic* -> Task Ar**i**thmetic

---

> ### Author Response · Authors · 2023-11-21
>
> Thank you very much for your review and affirmation of our work. We'll answer your questions one by one below.
>
> ## 1. About the question of "The experimental conclusion in Section 3.2.2 seems a bit strong":
>
> Based on your suggestions, we have added a correlation analysis of entropy and loss at different training stages. As shown in the table below, the Spearman correlation coefficient of entropy and loss is always high, indicating that **Entropy and Loss are positively correlated at different stages** , and entropy minimization can be used as an effective proxy for cross-entropy loss to optimize the merging coefficient of AdaMerging. Experimental results of Spearman correlation coefficients on eight datasets are added to the modified version of Figure 10.
>
> | Iterations | Spearman correlation coefficient |
> | :------: | :------: |
> | 0     | 0.87 |
> | 100   | 0.89 |
> | 200   | 0.91 |
> | 300   | 0.91 |
> | 400   | 0.92 |
> | 500   | 0.92 |
>
> ## 2. About the question of "The unrealistically accessible full test data":
>
> Yes, access to the full test dataset may not be practical. Based on your suggestions, we added how the performance of our method changes when different amounts (0.1%, 1%, 5%, 100%) of the test dataset can be accessed. As shown in the table below, we observe that our method also has an improvement of about 5% when only 0.1% of the test data is accessible. When 5% of the test data is accessible, our method achieves comparable accuracy to when the whole test data is available.
> | Method | Avg ACC (\%) | | Method | Avg ACC (\%) |
> | :------ | :------: |   :------: |   :------ | :------: |
> | Task Arithmetic (0%)  | 69.1 |    |  TIES-Merging (0%)  |  72.4 |
> | AdaMerging (0.1%)   | 74.0 |  |  AdaMerging++ (0.1%) | 77.9  |
> | AdaMerging (1%)   | 77.5 |  |  AdaMerging++ (1%) | 80.6  |
> | AdaMerging (5%)   | 80.1 |  |  AdaMerging++ (5%) |   81.0 |
> | AdaMerging (100%) | 80.1 |  |  AdaMerging++ (100%) |  81.1  |
>
> ## 3. About the question of "The ViT architecture and MTL baseline":
>
> In the experiment, we strictly followed the ViT architecture used by Task Arithmetic and Ties-Merging. Task Arithmetic, Ties-Merging and our AdaMerging did not require the training data of the original task, and only needed the provided trained models to construct the task vectors. Each task vector can be understood as the ability to complete a specific task. Furthermore, these task vectors are merged into the pre-trained model through merging coefficients to achieve MTL. Of course, our approach is model-agnostic and can be used with foundation models of any architecture.
>
> In addition, our method is compared fairly with baselines that merge models without original training data. As for more advanced MTL baselines such as GradNorm, PCGrad, GradDrop, etc., they are not the main baselines of the model merging community at this stage. Because there is still a gap between the current model merging and the traditional MTL baseline, these advanced MTL baselines need to be further considered only when this gap is further narrowed or even surpassed in the future. Of course, model merging technology has many advantages compared to traditional MTL methods. First, it does not require raw data for each task, significantly reducing the need for raw data collection and protecting data privacy. In addition, model merging is very efficient (especially for foundation models) compared with traditional MTL to train models from scratch, and it can usually be completed in minutes to tens of minutes.
>
> ## 4. About the question of "Overall summary":
>
> (1) We compared the current SOTA advanced task vector based MTL methods, Task Arithmetic (ICLR'2023), and Ties-Merging (NeurIPS'2023). Advanced MTL methods based on raw data training, such as GradNorm (ICML'2018), PCGrad (NeurIPS'20), GradDrop (NeurIPS'20), are not applicable baselines in model merging problem since they require original training data. Methods based on raw data are not our most relevant baseline for two reasons. On the one hand, there is still a performance gap between current SOTA model merging schemes and the naive MTL method based on raw data, so the "learning from models" [1] community is still working hard to fill this gap. On the other hand, traditional MTL based on raw data has many shortcomings compared with model merging. For example, model merging does not require efforts to collect a large amount of original data, and it also protects data privacy. More importantly, model merging is very efficient compared to training an MTL model from scratch, especially in the era of large models.
>
> (2) We further added correlation analysis (in Figure 10) between entropy and loss at different training stages, and we found that they are always highly correlated, which further supports our conclusion.
>
> [1] Zheng, Hongling, et al. "Learn From Model Beyond Fine-Tuning: A Survey." arXiv preprint arXiv:2310.08184 (2023).

---

> ### Author Response · Authors · 2023-11-21
>
> ## 5. About the question of "Reweighting/rebalancing traditional MTL model using task vector weights":
>
> Traditional MTL baseline does not use a weighting strategy. Task vector-based model merging focuses on model merging without original data, rather than training an MTL model from scratch. Therefore, if original data is available, it would be a better choice to directly use the original data to train an MTL model (including task/loss weighting coefficients). In other words, it is unlikely that the scenario of learning the merging coefficients of a task vector first and then using these coefficients to train an MTL model from scratch will occur.
>
> ## 6. About the question of "Supervised variant of AdaMerging":
>
> Thank you very much for your suggestion. We have added a supervised version of AdaMerging, that is, training AdaMerging directly with cross-entropy loss on the test dataset instead of entropy minimization. The experimental results are shown in the table below. We found that the unsupervised version of AdaMerging is very close to the supervised version of AdaMerging. This also shows from another perspective that it is reasonable for us to use entropy minimization as the proxy objective of cross-entropy loss to train the merging coefficients. In general, the performance gap between the task vector-based model merging methods and traditional MTL may be caused by a certain amount of information loss during the task vectors merging process.
>
> | Method | Label | ACC (%) |
> | :------ | :------ |  :------: |
> | Task Arithmetic  | - | 69.1 |
> | Task-wise AdaMerging       | Surpvised (upper bound) |71.3  |
> | Task-wise AdaMerging       | Unsurpvised (ours) | 71.1 |
> | Layer-wise AdaMerging      | Surpvised (upper bound) | 81.6 |
> | Layer-wise AdaMerging      | Unsurpvised (ours) | 80.1 |
>  |  |  |  |
> | TIES-Merging     | - | 72.4 |
> | Task-wise AdaMerging++     | Surpvised (upper bound) | 73.7 |
> | Task-wise AdaMerging++     | Unsurpvised (ours) | 73.7 |
> | Layer-wise AdaMerging++    | Surpvised (upper bound) | 82.0 |
> | Layer-wise AdaMerging++    | Unsurpvised (ours) | 81.1 |

---

> > ### Comment · Reviewer_3WDA · 2023-11-21
> > **Thanks for your response**
> >
> > Dear authors,
> > thanks a lot for your responses, and it's nice to see that AdaMerging still performs well with a limited test amount of data available.
> >
> > About your results from answer 6 (supervised variant of AdaMerging): Do you maybe have some insights of what this means with respect to MTL ? Intuitively, I would expect this method to perform very well as it can combines task vectors at a fine-grained level (layerwise) and with extra test data information. However it is still quite far from traditional MTL (88.9). Is it fair to interpret this as an inherent limit of task merging ?

---

> > > ### Author Response · Authors · 2023-11-22
> > >
> > > Dear reviewer, we have worked hard to address your question regarding the performance gap between model merging and traditional MTL. Do you still have questions? Looking forward to your reply.

---

> > > > ### Comment · Reviewer_3WDA · 2023-11-22
> > > > **Thanks for your response**
> > > >
> > > > Dear authors,
> > > > Thanks a lot for your response and  no, I do not have further questions.
> > > > Currently, I am inclined to keep my rating. While I do see clear positives in the paper, especially when compared to traditional task merging, I am still on the fence about the novelty/strength of the contribution and where exactly to place it in the literature.
> > > >
> > > > To be more specific, I understand that the paper focuses on task merging as baselines; however, from my understanding, the proposed method loses some of the practical benefits of standard task merging as it requires (unlabeled) test data and additional training time. I think these changes make traditional MTL into a more relevant baseline than it is for traditional task merging, and introduces different trade-offs. For instance:
> > > >   - **Data:** MTL requires labelled training data while AdaMerging requires unsupervised test samples; both of these assumptions can seem unrealistic/impractical depending on the setting.
> > > >  -  **Compute:** Both of them requires training time, although I assume the exact comparison will depend on the training setup of MTL (e.g. parameter-efficient finetuning, full finetuning, training from scratch, etc)
> > > >  - **Memory:** Memory wise, MTL only requires a single model; while AdaMerging requires storing and executing one pretrained model for each task when optimizing the merging coefficients
> > > > - **Generalization:** As pointed out in the paper (Table 3 and 4), generalization/robustness to new tasks is an important property of MTL. However AdaMerging assumes the availability of (unlabelled) test samples, which seems to be a different setup than generalizing to an entirely unseen test set.

---

> > > > > ### Author Response · Authors · 2023-11-23
> > > > >
> > > > > Dear reviewer, thank you for your reply. Firstly, it's crucial to distinguish between model merging and traditional Multi-Task Learning (MTL) as two inherently different multi-task solutions tailored for distinct settings. When original task training data is accessible, opting for the traditional MTL scheme seems the most intuitive decision. However, if the original training data is absent, the conventional MTL approach becomes impractical and inapplicable. In such instances, model merging emerges as a viable alternative. **Our paper concentrates on the nuanced challenge of executing MTL without the presence of training data, a scenario where traditional MTL is not applicable. It's worth noting that employing traditional MTL with training data merely establishes a performance upper bound.**
> > > > >
> > > > >
> > > > > Regarding the replies to several questions you mentioned:
> > > > >
> > > > > **Data**: Whether operating within the framework of model merging or traditional MTL, once a pre-trained model is deployed in a real-world scenario, it will inevitably encounter test data; otherwise, the model has no real value. Consequently, we assert that the assumption of the availability of unlabeled test data is valid and imposes minimal requirements. Our experiments provide additional evidence that AdaMerging maintains its effectiveness even when only 0.1% of the (unlabeled) test data is accessible—equivalent to just a few to dozens of samples for each task. This showcases the remarkable data efficiency of AdaMerging. In contrast, it's important to note that traditional MTL is not applicable when the original training data is unavailable, which is the primary focus of our paper.
> > > > >
> > > > >
> > > > >
> > > > > **Compute**: AdaMerging demonstrates its superior efficiency in terms of data, parameters, and training time. To illustrate, AdaMerging excels with a minimal requirement of 0.1%-5% of unlabeled test data. Task-wise AdaMerging consists of only 8 parameters, while the Layer-wise version consists of only 1,248 parameters. Notably, significant performance enhancements are achieved within a brief training period, ranging from a few minutes to ten minutes. In contrast, traditional MTL approaches such as parameter-efficient fine-tuning, full fine-tuning, and training from scratch prove impractical without access to the original task training data—an aspect central to our paper's focus. AdaMerging, on the other hand, circumvents this limitation without needing the original training data.
> > > > >
> > > > >
> > > > >
> > > > > **Memory**：During the training of model merging coefficients, AdaMerging requires the storage of the task vector for each task, the same as the SOTA methods, including Task Arithmetic [1] and Ties-Merging [2]. Notably, AdaMerging does not necessitate the storage of training data for individual tasks during this phase. In addition, once the coefficients are acquired, AdaMerging only retains a single consolidated model for deployment or testing, incurring a deployment cost identical to that of traditional MTL. In contrast, while traditional MTL only requires the storage of a single model, it mandates the retention of the original training data for each task. The size of task-specific training data can be substantial, potentially surpassing the model size in practical real-world applications.
> > > > >
> > > > >
> > > > > **Generalization**: The setup of generalization/robustness here is slightly different from the traditional MTL setting with original training data. It's crucial to highlight that our work aligns with the research trajectory of task arithmetic [1] and Ties-Merging [2], specifically without original task training data, as opposed to the conventional MTL scenario which heavily relies on task training data. Furthermore, our comparison with the baselines, particularly Task Arithmetic in Tables 3 and 4, remains fair. Task Arithmetic involves the manual merging of model weights, determined through a search process using labeled validation data. Consequently, the outcomes presented in Tables 3 and 4 effectively demonstrate that AdaMerging substantially surpasses Task Arithmetic in terms of generalization and robustness under fair comparisons.
> > > > >
> > > > >
> > > > > In summary, **while traditional MTL and model merging each possess different computational and memory advantages, they represent distinct technological choices tailored to entirely different settings. In situations where original training data is absent, traditional MTL becomes impractical and inapplicable, leaving the model merging approach as the only viable option**. AdaMerging stands out with superior performance compared to other SOTA model merging baselines, such as Task Arithmetic [1] and Ties-Merging [2].
> > > > >
> > > > >
> > > > >
> > > > > Reference:
> > > > >
> > > > > [1] Editing models with task arithmetic. ICLR 2023
> > > > >
> > > > > [2] TIES-MERGING: Resolving Interference When Merging Models. NeurIPS 2023

---

> > > > > ### Author Response · Authors · 2023-11-23
> > > > >
> > > > > Dear reviewer, model merging and traditional MTL are two multi-task learning solutions under different settings, mainly selected based on whether the original data is available. We have further explained your questions. Since only a few hours are left in the rebuttal phase, we look forward to getting your latest assessment. Thanks.

---

> > > > > > ### Comment · Reviewer_3WDA · 2023-11-23
> > > > > > **Thanks for your response**
> > > > > >
> > > > > > Dear authors,
> > > > > >
> > > > > > Thanks for your added response, and let me clarify that I do agree that MTL and task merging are two different settings. In all fairness, I do not expect AdaMerging and traditional MTL to be 1-to-1 compared. However, in a similar way, I also disagree that comparing traditional task merging and AdaMerging is a 100% fair comparison: In my opinion, AdaMerging introduces a unique experimental setup in between traditional task merging and traditional MTL, and I do not think the current writing fully reflects this, as it focuses mainly on task merging as baselines.
> > > > > >
> > > > > > **once a pre-trained model is deployed in a real-world scenario a real-world scenario, it will inevitably encounter test data**:
> > > > > > I agree with this; However, a real-world scenario is more likely to follow a continual/online learning setup where the test data used to train the merging coefficient will be different from the test data the model is actually tested on later. In contrast (to my understanding) the main experimental setup of AdaMerging evaluates on the same test data that is used to learn the coefficients.
> > > > > >
> > > > > > **Generalization:** I respectfully disagree that the comparison to task merging is perfectly fair there. First that having access to the unlabelled test we want to generalize on does not match the usual definitions of generalization or robustness. And secondly, while it is true "Editing models with task arithmetic" uses extra data, according to their original paper, it uses "held-out validation sets", which is  different from having access to unlabelled test data.
> > > > > >
> > > > > > **In situations where original training data is absent, traditional MTL becomes impractical and inapplicable, leaving the model merging approach as the only viable option**:
> > > > > > I think this point is a good illustration for what I mean with *AdaMerging introduces a unique experimental setup in between traditional task merging and traditional MTL*. In the AdaMerging scenario we have access to *(i)* a pretrained generic network, *(ii)* unlabeled  test data and *(iii)* prefinetuned single task networks. With these assumptions, traditional MTL seems applicable to me e.g. via few-shot distillation (for instance similar to **[1]**), or generating proxy labels with the single-task networks for supevised finetuning.
> > > > > > And to be fair, I do not think these methods should necessarily be used as baselines, but I think they should at least be accounted for and discussed in the paper, rather than describing traditional MTL as an entirely different/inapplicable solution.
> > > > > >
> > > > > >
> > > > > > **[1]** Universal Representations: A Unified Look at Multiple Task and Domain Learning, Li et al, 2022

---

> ### Author Response · Authors · 2023-11-22
>
> Dear reviewer, thanks for your time and response!
>
> I think there are many possible reasons why there is still a gap between the current model merger and traditional MTL. This is mainly because the models for each task are trained in isolation, that is, they may use different learning rates, number of epochs, and other different hyperparameters to train the models. Naturally, the task vectors constructed for each task have many inherent differences, so there is inevitable information loss when merging modular task vectors.
>
> For example, some tasks may converge to a sharp local minimum [1], that is, a slight perturbation of the weights may cause a significant change in the loss. Therefore, when the model is merged, the performance of this part of the task will decrease significantly, thereby affecting the overall performance. In addition, the local minimum points that different tasks converge to may be in different basins [2], and their weights may need to be rearranged and combined before merging for alignment. Finally, the size of the merged architecture also has a great impact on performance. For example, when merging larger architectures (they usually have better cross-task generalization), the performance of task vector based methods will be closer to that of traditional MTL. As shown in the table below, when merging ViT-L/14 (342,562,049 parameters per task vector) and ViT-B/32 (113,448,705 parameters per task vector), the gap is different. On ViT-L/14, AdaMerging++ can achieve an accuracy of 91.0%, which is very close to the 93.5% of traditional MTL.
>
> | Method | Architecture | Average Accuracy (%) |
> | :------ | :------: | :------: |
> | Task Arithmetic      | ViT-B/32  | 69.1 |
> | Ties-Merging         | ViT-B/32 | 72.4 |
> | AdaMerging (unsupervised)          | ViT-B/32 | 80.1 |
> | AdaMerging++ (unsupervised)        | ViT-B/32 | 81.1 |
> | Traditional MTL      | ViT-B/32 | 88.9 |
> |||
> | Task Arithmetic      | ViT-L/14 | 84.5 |
> | Ties-Merging         | ViT-L/14 | 86.0 |
> | AdaMerging (unsupervised)          | ViT-L/14 | 90.8 |
> | AdaMerging++ (unsupervised)        | ViT-L/14 | 91.0 |
> | Traditional MTL      | ViT-L/14 | 93.5 |
>
>
> In the future, we will explore these directions more deeply to further narrow the gap in model merging. **This paper focuses on reducing this gap from the perspective of merging coefficients**.
>
> [1] Foret, Pierre, et al. "Sharpness-aware minimization for efficiently improving generalization." ICLR 2021.
>
> [2] Ainsworth, Samuel K., Jonathan Hayase, and Siddhartha Srinivasa. "Git re-basin: Merging models modulo permutation symmetries." ICLR 2023.

---

> ### Author Response · Authors · 2023-11-23
>
> Dear Reviewer,
>
> Thanks for your response.
>
> **Regarding the fair comparisons**:
>
> According to your suggestions, **we conducted experiments on a held-out validation set without access to the unlabeled test data, using the identical setup as Task Arithmetic [1] and TIES-Merging [2]. This approach ensures a fair comparison with the model merging baselines [1,2] mentioned.**.  The results are shown in the following table (Due to time constraints, we only had time to perform 100 iterations, and the complete results will be updated later.).
>
> | Method | SUN397 | Cars | RESISC45 | EuroSAT | SVHN | GTSRB | MNIST | DTD | Avg Acc|
> | :------ | :------: |  :------: |:------: |  :------: |:------: |  :------: |:------: |  :------: | :------: |
> | Task Arithmetic   |  55.2 | 54.9  | 66.7  | 78.9 |  80.2 |  69.7 |  97.3 |  50.4  | 69.1 |
> | AdaMerging("unlabled test sets", 100-iteration) | 59.9 | 60.9 | 72.8 | 90.2 | 84.1 | 76.8 | 97.6 | 52.0 | 74.3 |
> | AdaMerging("held-out validation sets", 100-iteration) | 59.8 | 61.4 | 72.8 | 90.4 | 83.6 | 76.4 | 97.6 | 53.4 | 74.4 |
>
>
> | Method | Avg ACC of seen tasks (SUN397, Cars, RESISC45, SVHN, GTSRB, DTD) |  Avg ACC of unseen tasks (MNIST, EuroSAT) |
> | :------ | :------: |  :------: |
> | Task Arithmetic   |  70.6  |    61.7  |
> | AdaMerging("unlabled test sets", 100-iteration) |  73.7  |  64.9  |
> | AdaMerging("held-out validation sets", 100-iteration) |  73.8   |  63.2  |
>
>
> **In conclusion, by employing the same setup and ensuring the same and fair comparisons with model merging baselines [1,2], we demonstrate that utilizing the held-out validation dataset, without leveraging the unlabeled test data for coefficient tuning, yields comparable performance results.**
>
>
> Reference:
>
> [1] Editing models with task arithmetic. ICLR 2023
>
> [2] TIES-MERGING: Resolving Interference When Merging Models. NeurIPS 2023

---

> > ### Comment · Reviewer_3WDA · 2023-11-23
> > **Thanks for the reponse**
> >
> > Dear authors,
> > Thanks a lot for the quick reply and the proof of concept experiment. Taking into account the updated results, I am increasing my score to 6 as my main concern is addressed, and I'm hoping the trade-offs of test data requirement and related experiments (held-out validation data + results with limited amount of test data) can be discussed more in-depth in the paper.

---

### Official Review · Reviewer_88YY · 2023-11-01

**Soundness:** 3 good
**Presentation:** 4 excellent
**Contribution:** 3 good
**Rating:** 8
**Confidence:** 4

**Summary:**

The paper first points out that multi-task model merging coefficients have a significant impact on the performance of existing merging solutions, and grid search model merging coefficients are unrealistic. Then, this paper proposes a new merging scheme, AdaMerging, based on multi-task entropy minimization to learn the optimal merging coefficients. Finally, the results under various architectures show that the proposed solution is significantly improved compared to existing model merging solutions.

**Strengths:**

1. This paper studies model merging without original data, which is an important research direction.
2. This paper proposes an unsupervised model merging scheme, which is technically feasible. Experimental results show that the proposed scheme has better multi-task performance, generalization, and robustness.
3. The paper is well organized and easy to understand, and the proposed solutions are easy to follow and implement.

**Weaknesses:**

1. In the motivation, the authors need to explain the intuitive motivation for entropy minimization as a proxy objective for loss.
2. In the experimental analysis, the author needed to explain why AdaMerging has better generalization and robustness.

**Questions:**

1. Is AdaMering in Tables 2, 3, and 4 a Task-wise or a Layer-wise version?
2. Why is the model merging performance closer to traditional MTL in a larger architecture?

---

> ### Author Response · Authors · 2023-11-21
>
> Thank you very much for your review and affirmation of our work. We'll answer your questions one by one below.
>
> ## 1. About the question of "Intuitive motivation for entropy minimization":
>
> The intuitive motivation for entropy minimization is to reduce model uncertainty. The lower the Shannon entropy, the more concentrated the probability distribution is and the more confident the model's predictions are. By minimizing entropy, we want the model to produce a more deterministic output when faced with a given input, resulting in a more robust and accurate model.
>
> ## 2. About the question of "Generalization and robustness of AdaMerging":
>
> Our AdaMerging allows us to adapt to unlabeled test data of unseen tasks or unlabeled corrupted data in an unsupervised way when training model merging coefficients, thereby obtaining model merging coefficients that are most suitable for these data. Therefore, the AdaMerging merged model better bridges the gap between training and testing data distribution shifts. However, existing methods ignore this point. Therefore, our method will have better generalization and robustness compared with existing task vector-based model merging methods.
>
> ## 3. About the question of "AdaMerging version":
>
> AdaMerging in Tables 2, 3, and 4 are all Layer-wise versions. As we introduced in the experimental settings in Section 4.1, all versions not specifically emphasized are Layer-wise.
>
> ## 4. About the question of "The relationship between model architecture and performance":
>
> On the one hand, larger architectures usually have more parameters and more powerful representation learning capabilities, and larger architectures can better learn shared feature representations. These shared underlying features benefit knowledge transfer between different tasks, so the merged model performs better and is closer to traditional MTL. On the other hand, larger architectures have more layers, and our Layer-wise AdaMerging can have more learnable merging coefficients. For example, under ViT-B/32 and ViT-L/14, the number of coefficients for Layer-wise AdaMerging is 1,248 and 2,408, respectively. As a result, we can balance the task vectors more flexibly, leading to better performance.

---

### Meta-Review · Area_Chair_vjke · 2023-12-08

**Metareview:**

This paper studies the recently popularized problem of combining multiple single-task models (that are fine-tuned from the same pre-trained checkpoint) into a single multitask model. Many past works make use of hyperparameters that determine how to weight the contribution of each individual model; this paper proposes optimizing an entropy minimization loss (common in the semi-supervised learning literature) on unlabeled test examples to choose these hyperparameters. The resulting method, AdaMerging, outperforms prior merging methods in terms of multitask performance in standard merging benchmark settings. The primary drawback is the assumption of a set of unlabeled test datapoints, which was not an assumption that was made by past work (though some past work assumed access to *labeled* validation datapoints, which is probably even more restrictive). The other primary weakness is that the paper relies heavily on an trivial connection between entropy and cross-entropy. Beyond that, I would point out that the naming is a bit odd - AdaMerging++ is AdaMerging applied to TIES merging, but you can't really tell that just from the "++". Something like TIES-AdaMerging may be clearer.

**Justification For Why Not Higher Score:**

While all reviewers agreed it should be accepted, the reliance on unlabeled held-out data is a meaningful limitation, and the contribution on the whole is not dramatic.

**Justification For Why Not Lower Score:**

All reviewers agreed on acceptance.

---

### Decision · Program_Chairs · 2024-01-16

Accept (poster)